# Enhancing Efficiency of Safe Reinforcement Learning via Sample Manipulation

**Shangding Gu**[1,3,*] **Laixi Shi**[2*] **, Yuhao Ding**[1]**, Alois Knoll**[3]**, Costas Spanos**[1]**, Adam Wierman**[2]**,**
**Ming Jin**[4]
[1]University of California, Berkeley, USA
[2]California Institute of Technology, USA
[3]Technical University of Munich, Germany
[4]Virginia Tech, USA

## Abstract

Safe reinforcement learning (RL) is crucial for deploying RL agents in real-world applications, as it aims to maximize long-term rewards while satisfying safety constraints. However, safe RL often suffers from sample inefficiency, requiring extensive interactions with the environment to learn a safe policy. We propose Efficient Safe Policy Optimization (ESPO), a novel approach that enhances the efficiency of safe RL through *sample manipulation*. ESPO employs an optimization framework with three modes: maximizing rewards, minimizing costs, and balancing the trade-off between the two. By dynamically adjusting the sampling process based on the observed conflict between reward and safety gradients, ESPO theoretically guarantees convergence, optimization stability, and improved sample complexity bounds. Experiments on the *Safety-MuJoCo* and *Omnisafe* benchmarks demonstrate that ESPO significantly outperforms existing primal-based and primal-dual-based baselines in terms of reward maximization and constraint satisfaction. Moreover, ESPO achieves substantial gains in sample efficiency, requiring 25–29% fewer samples than baselines, and reduces training time by 21–38%.

## 1   Introduction

Reinforcement learning (RL) [49] has demonstrated powerful capabilities in various domains [25, 46, 11]. However, ensuring safety in RL, particularly in real-world applications such as autonomous driving, robotics, and power grids, is crucial [6, 12, 31, 35, 38, 39, 41, 48, 55, 57, 17, 59]. Safe RL aims to maximize long-term cumulative rewards while adhering to additional safety cost constraints.

Most state-of-the-art (SOTA) safe RL methods, including primal-based baselines (e.g., CRPO [53], PCRPO [30]) and primal-dual-based methods (e.g., CUP [54], PPOLag [32]), optimize cost and reward with a predetermined sample size for each iteration. However, this approach may lead to *sample inefficiency* due to two main reasons:

- ❚ Wasted samples and computational resources in simple scenarios, where the (computational/physical) cost of obtaining these samples may outweigh their learning benefits.
- ❚ Insufficient exploration in complex cases with high uncertainty or conflicting objectives, potentially hindering the learning of a safe and optimal policy.

A key insight from optimization literature suggests that the selection of sample size is worthwhile but a delicate issue, as it may vary depending on the optimization stage and landscape [13, 28, 51]. However, this insight remains largely unexplored in the context of safe RL, where the consideration

---

*Equal contribution.

of safety adds complexity. The presence of safety constraints can create regions with high conflict between reward and safety objectives, requiring careful balancing and potentially more samples to resolve. Therefore, an unresolved question in safe RL is: **Can we enhance sample efficiency by dynamically adapting the sample size, while simultaneously improving reward performance and guaranteeing safety?**

To address this question, we focus on primal-based approaches, which do not require fine-tuning of dual parameters or heavily rely on initialization, unlike primal-dual-based optimization [30, 53]. The key to effectively enhancing sample efficiency is to establish reliable criteria for determining sample size requirements. Inspired by insights from multi-objective optimization/RL [40, 37, 30], we use *gradient conflict between rewards and costs* as an effective signal for adjusting sample size in each iteration. Intuitively, when gradient conflict occurs, balancing reward and safety optimization with a uniform sample size becomes challenging; conversely,

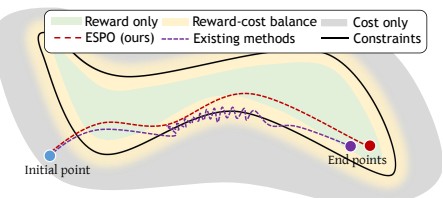

Figure 1: Oscillation analysis compared our method with existing safe RL methods in three modes of optimization.

when there is gradient alignment, optimizing with fewer samples is more straightforward. This motivates us to adopt a three-mode optimization framework: 1) optimizing cost exclusively upon a safety violation; 2) simultaneously optimizing both reward and cost during a soft constraint violation; 3) optimizing only the reward when no violations are present. This allows tailored sample size adjustment based on the optimization regime. We increase the sample size in situations of gradient conflict to incorporate more informative samples and reduce it in cases of gradient alignment to prevent unnecessary costs and training time. This sampling adjustment is effective in each policy learning mode (cost only, simultaneous reward and cost, and reward only), enabling the search for improved policies that prioritize safety, rewards, or a balance of both.

This study makes three key contributions emphasizing sample manipulation for safe RL:

① We propose Efficient Safe Policy Optimization (ESPO), an algorithm that depart from prior arts byincorporating sample manipulation by leveraging gradient conflict signals as criteria to enhance sample efficiency and reduce unnecessary interactions with the environments, . ② We provide a comprehensive theoretical analysis of ESPO, including convergence rates, the advantages of reducing optimization oscillation, and provable sample efficiency. The theoretical results inspire ESPO's sample manipulation approach and could be of independent interest for broad RL applicability. ③ We evaluate ESPO through comparative and ablation experiments on two benchmarks: *Safety-MuJoCo*[30] and *Omnisafe*[32]. The results demonstrate that ESPO improves reward performance and safety compared to SOTA primal-based and primal-dual-based baselines. Notably, ESPO significantly reduces the number of samples used during policy learning and minimizes training costs while ensuring safety and achieving superior reward performance.

## 2 Related Works

Various methodologies have been developed to enhance safety in RL [12, 31], including constrained optimization-based methods, control-based methods [18, 19, 33, 29], and formal methods [43]. Among these, constrained optimization-based methods have gained notable popularity due to their ease of use and reduced dependency on external knowledge [31].

Constrained optimization-based methods can be categorized into primal-dual (e.g., CPO [1], PCPO [56], CUP [54]) and primal approaches. Primal-dual methods face challenges in tuning dual multipliers, ensuring feasible initialization, and sensitivity to learning rates [53, 30]. Primal methods offer a distinct advantage by eliminating the need for dual multipliers. A prominent primal-based method is CRPO [53], which focuses on directly optimizing the primal problem. When safety violations occur, CRPO exclusively improves the violated constraints. However, it encounters significant challenges with conflicting gradients between optimizing rewards and constraints, which can impact ensuring both performance and ongoing safety compliance. PCRPO [30] addresses this issue by balancing the trade-offs between reward and safety performance through strategic gradient manipulation. However, it lacks comprehensive convergence and sample complexity analysis and faces computational challenges due to the need to compute reward and safety gradients in each gradient handling step.

Several efficient safe RL methods have been recently proposed [16, 21, 22, 23, 24, 36, 42, 47, 50], including offline [47] and off-policy settings [36, 42]. Our model-free, on-policy approach is distinguished by its dynamic calibration of sampling based on the interplay between reward maximization and safety assurance. Closely related works are [21] and [24]. [21] employs symbolic reasoning for safety but relies on external knowledge, potentially limiting applicability. [24] proposes a non-stationary safe RL approach with regret bounds using linear function approximation but may struggle with complex tasks and inherits issues common in primal-dual safe RL [22, 23]. Our primal-based method circumvents these drawbacks.

Adaptive sampling methods in optimization can be categorized into prescribed (e.g., geometric) sample size increase [13, 26] [13, 8], gradient approximation test [14, 7, 13, 8, 15, 10, 5], and derivative-free [45, 9] and simulation-based methods [44] (see [20] for a review). These methods focus on controlling the variance of gradient approximations or function evaluations (e.g., through inner product [8] or norm tests [14, 15]) to balance computational efficiency and sample complexity. Adaptive sampling methods have also been applied to constrained stochastic optimization problems with convex feasible sets [4, 52]. A recent work [60] extends adaptive sampling to a multi-objective setting, but their criteria are still based on variance. Our research introduces a novel perspective by focusing on conflict-aware updates based on safety and performance gradients in safe RL, making it the first adaptive sampling method for this important domain.

## 3    Problem Formulation

A Constrained Markov Decision Process (CMDP) [3] is often used to model safe RL problems. A CMDP is denoted as $(\mathcal{S}, \mathcal{A}, P, r, c, b, \gamma)$, where $\mathcal{S}$ is the state space, $\mathcal{A}$ is the action space, $P : \mathcal{S} \times \mathcal{A} \times \mathcal{S} \to [0, 1]$ is the transition probability function, $r : \mathcal{S} \times \mathcal{A} \to \mathbb{R}$ is the reward function, and $\gamma$ is the discount factor. To encode safety, $c = (c_1, \ldots, c_n) : \mathcal{S} \times \mathcal{A} \to \mathbb{R}^n$ is the cost function assigning costs to state-action pairs, with higher costs indicating higher risks, $b = (b_1, \ldots, b_n) \in \mathbb{R}^n$ contains safety thresholds for each constraint. This CMDP framework searches for a safe policy $\pi$ in the stochastic Markov policy set $\Pi$, balancing rewards and safety constraints.

The expected cumulative reward values are defined as $V_r^\pi(s) = \mathbb{E}\left[\sum_{t=0}^{\infty} \gamma^t r(s_t, a_t) \middle| \pi, s_0 = s\right]$ and $Q_r^\pi(s, a) = \mathbb{E}\left[\sum_{t=0}^{\infty} \gamma^t r(s_t, a_t) \middle| \pi, s_0 = s, a_0 = a\right]$ for states and state-action pairs, respectively. Similarly, safety is quantified using the cost state values $V_c^\pi(s)$ and cost state-action values $Q_c^\pi(s, a)$. The primary objective in safe RL is to maximize the accumulative reward while ensuring safety, under an initial state distribution $\rho$:

$$\max_{\pi \in \Pi} V_r^\pi(\rho) \coloneqq \mathbb{E}_{s \sim \rho}\left[V_r^\pi(s)\right], \ \text{ s.t. } V_c^\pi(\rho) \coloneqq \mathbb{E}_{s \sim \rho}[V_c^\pi(s)] \le b. \tag{1}$$

However, conflicts often arise in safe RL between the reward gradient $\mathbf{g}_r = \nabla V_r^\pi(\rho)$ and negative cost gradient $\mathbf{g}_c = -\nabla V_c^\pi(\rho)$. These conflicts can lead to unstable policy updates that cause experiences violating safety constraints, forcing reversion to prior policies and wasting samples. Such unstable dynamics further impede efficient exploration, risking premature convergence and squandering of computational resources. This study aims to efficiently search for a safe policy by manipulating samples to reduce waste and improve safe RL efficiency.

## 4    Algorithm Design and Analysis

### 4.1    Three-Mode Optimization

To improve learning efficiency and mitigate oscillations, we leverage PCRPO [30] and categorize performance optimization into three distinct strategies: focusing on reward, on both reward and cost simultaneously, or solely on cost. Two essential parameters are introduced to construct a soft constraint region — $h^-$ on the lower side and $h^+$ on the upper side. With $h^-, h^+$ in hand, [30] divides the optimization process into three modes as below. Throughout the paper, we parameterize the policy $\pi$ by $w$.

• **1) Safety Violations.** When the cost values $V_c^\pi(\rho) > (h^+ + b)$, we apply (2) to update the policy parameter $w_t$ with learning rate $\eta$. In such mode, since the constraints are violated, we prioritize

safety and choose to minimize the cost objective to achieve compliance with safety standards.

$$w_{t+1} = w_t + \eta \mathbf{g}_c. \tag{2}$$

• **2) Soft Constraint Violations.** When $V_c^\pi(\rho) \in [h^- + b, h^+ + b]$, we leverage (3) and (4) for simultaneous optimization of reward and safety performance. Specifically, when within the soft constraint region, the *conflict* between the reward and cost gradients is characterized by the angle $\theta_{r,c}$ between the reward gradient $\mathbf{g}_r$ and the cost gradient $\mathbf{g}_c$. When $\theta_{r,c} > 90°$, it indicates the directions that optimize the reward and the safety performance are in conflict, and the update rule is (3).

$$w_{t+1} = \begin{cases} w_t + \eta \left[ x_t^r \left( \mathbf{g}_r - \dfrac{\mathbf{g}_r \cdot \mathbf{g}_c}{\|\mathbf{g}_c\|^2} \mathbf{g}_c \right) + x_t^c \left( \mathbf{g}_c - \dfrac{\mathbf{g}_c \cdot \mathbf{g}_r}{\|\mathbf{g}_r\|^2} \mathbf{g}_r \right) \right], & \text{(3)} \\ w_t + \eta \left[ x_t^r \mathbf{g}_r + x_t^c \mathbf{g}_c \right], & \text{(4)} \end{cases}$$

where $x_t^r, x_t^c \geq 0$ and $x_t^r + x_t^c = 1$ for all $t \in T$. It employs gradient projection techniques [30, 58], projecting reward and cost gradients onto their normal planes and ensuring that the policy adjustment balances the conflicting objectives of maximizing rewards and minimizing costs. In contrast, when $\theta_{r,c} \leq 90°$, namely, the directions for maximizing rewards and minimizing costs are aligned or do not significantly oppose each other, we use the update rule (4). In this scenario, the gradient for the update is computed based on the weight of the reward and cost gradients. This method leverages the synergistic potential between reward maximization and cost minimization, aiming for a policy update that harmoniously improves both aspects.

• **3) No Violations.** When $V_c^\pi(\rho) < (h^- + b)$, the update rule in (5) is applied to optimize the policy:

$$w_{t+1} = w_t + \eta \mathbf{g}_r. \tag{5}$$

In other words, given that the policy adheres to all specified constraints, only the reward objective is considered.

## 4.2  Sample Size Manipulation

As introduced above, PCRPO [30] allows for adaptive optimization updates based on different conditions. However, PCRPO and other existing safe RL methods usually apply an identical sample size during the learning process, resulting in potentially unnecessary computation cost for simpler tasks and inadequate exploration for more complex tasks. Furthermore, there is no existing theoretical analysis for PCRPO, leaving the performance guarantees of it somewhat uncharted. To address the above challenges, we propose a method called ESPO based on a crucial sample manipulation approach that will be introduced momentarily. A comprehensive theoretical analysis of ESPO is provided in Section 4.4.

Throughout the framework of three-mode optimization, our proposed method dynamically adjusts the number of samples utilized at each iteration based on the criteria of gradient conflict, to meet specific demands of reducing unnecessary samples in simpler scenarios and increasing exploration in more complex situations. Specifically, we consider the three-mode optimization classified by the gradient-conflict criteria respectively. **2)(a)** *Soft Constraint Violations with Gradient Conflict*, where $\theta_{r,c} > 90°$ (cf. (6)): the cases with slight safe constraint violation and gradient conflict between reward and safety objectives. In this scenario, adjusting the sample size becomes crucial for sufficiently exploring the environments to identify a careful balanced udpate direction. We increase the sample size in (6) to enhance the likelihood of achieving a near-optimal balance between the reward and cost objectives. **2)(b)***Soft Constraint Violations without Gradient Conflict*, where $\theta_{r,c} \leq 90°$ (cf. (7)): the cases with slight safe constraint violation and gradient alignment between reward and safet objectives. Considering it is easier to search for a update direction that benefits the aligned reward and cost objectives, we reduce the sample size in (7) to achieve efficient learning. **1) and 3)** *Safety Violations* and *No Violations*: only reward or cost objective is considered. It indicates that there is no gradient conflict since only one objective is targeted, where we also employ the update rule in (7).

For more details, we dynamically adjust the sample size $X_t$ ($X$ denote a default fixed sample size), with $\zeta_t^+$ and $\zeta_t^-$ representing some sample size adjustment parameters.

$$X_{t+1} = \begin{cases} X + X\zeta_t^+, & \text{if } \theta_{r,c} > 90°, & \text{(6)} \\ X + X\zeta_t^-, & \text{if } \theta_{r,c} \leq 90°. & \text{(7)} \end{cases}$$

This gradient-conflict-based sample manipulation is a crucial feature of our proposed method, which enables adaptively sample size tailored to the specific nature of the joint reward-safety objective landscape at each update iteration.

## 4.3 Efficient Safe Policy Optimization (ESPO)

Building upon the above two modules — three-mode optimization and sample size manipulation, we have formulated a practical algorithm. The details of this algorithm are summarized in Algorithm 1 in Appendix B. This algorithm encompasses a strategic approach to sample size adjustment and policy updates under various conditions: **1)** *Safety Violations*: When a safety violation occurs, we adjust the sample size $X_t$ using Equation (7). Simultaneously, the policy $\pi_{w_t}$ is updated to ensure safety, as dictated by Equation (2). **2)(a)** *Soft Constraint Violations with Gradient Angle* $\leq 90°$: In modes of soft region violation where the angle $\theta_{r,c}$ between gradients $\mathbf{g}_r$ and $\mathbf{g}_c$ is less than or equal to $90°$, we adjust the sample size $X_t$ using Equation (7). The policy $\pi_{w_t}$ is then updated in accordance with Equation (3). **2)(b)** *Soft Constraint Violations with Gradient Angle* $> 90°$: Conversely, if the soft region violation occurs with a gradient angle $\theta_{r,c}$ exceeding $90°$, the sample size $X_t$ is adjusted via Equation (6). Policy updates are made using Equation (4). **3)** *No Violations*: In the absence of any violations, the sample size $X_t$ is altered using Equation (7). The policy $\pi_{w_t}$ is then updated to maximize the reward $V_r^\pi(\rho)$, following Equation (5). This practical algorithm reflects an insightful analysis of the interplay between reward maximization and safety assurance in safe RL, tailoring the learning process to the specific demands of each scenario.

## 4.4 Theoretical analysis of ESPO

In this section, we provide theoretical guarantees for the proposed ESPO, including the convergence rate guarantee and provable optimization stability and sample complexity advancements.

**Tabular setting with softmax policy class.** In this paper, we focus on a fundamental tabular setting with finite state and action space. We consider the class of policies with the softmax parameterization which is complete including all stochastic policies. Specifically, a policy $\pi_w$ associated with $w \in \mathbb{R}^{|\mathcal{S}||\mathcal{A}|}$ is defined as

$$\forall (s,a) \in \mathcal{S} \times \mathcal{A}: \quad \pi_w(a|s) := \frac{\exp(w(s,a))}{\sum_{a' \in \mathcal{A}} \exp(w(s,a'))}. \tag{8}$$

Before proceeding, we introduce some useful notations. When executing ESPO (cf. Algorithm 1), let $\mathcal{B}_r$, $\mathcal{B}_{\mathsf{soft}}$, and $\mathcal{B}_c$ denote the set of iterations using *Safety Violation Response* (mode 1), *Soft Constraint Violation Response* (mode 2), and *No Violation Response* (mode 3) in Section 4.3, respectively.

**I: Provable convergence of ESPO.** First, we present the convergence rate of our proposed ESPO in terms of both the optimal reward and the constraint requirements in the following theorem; the proof is given in Appendix A.3.

**Theorem 4.1.** *Consider tabular setting with policy class defined in* (8)*, and any* $\delta \in (0,1)$*. For Algorithm 1, applying* $T_{\mathsf{pi}} = \widetilde{O}\big(\frac{T \log(\frac{|\mathcal{S}||\mathcal{A}|}{\delta})}{(1-\gamma)^3 |\mathcal{S}||\mathcal{A}|}\big)^2$ *iterations for each policy evaluation step, set tolerance* $h^+ = \widetilde{O}\big(\frac{2\sqrt{|\mathcal{S}||\mathcal{A}|}}{(1-\gamma)^{1.5}\sqrt{T}}\big)$ *and the learning rate of NPG update* $\eta = (1-\gamma)^{1.5}/\sqrt{|\mathcal{S}||\mathcal{A}|T}$*. Then, the output* $\widehat{\pi}$ *of Algorithm 1 satisfies that with probability at least* $1 - \delta$*,*

$$V_r^{\pi^\star}(\rho) - \mathbb{E}[V_r^{\widehat{\pi}}(\rho)] \leq \widetilde{O}\left(\sqrt{\frac{|\mathcal{S}||\mathcal{A}|}{(1-\gamma)^3 T}}\right), \quad \mathbb{E}[V_c^{\widehat{\pi}}(\rho)] - V_c^{\pi^\star}(\rho) \leq \widetilde{O}\left(\sqrt{\frac{|\mathcal{S}||\mathcal{A}|}{(1-\gamma)^3 T}}\right).$$

*Here, the expectation is taken with respect to the randomness of the output* $\widehat{\pi}$*, which is randomly selected from* $\{\pi_{w_t}\}_{1 \leq i \leq T}$ *with a certain probability distribution (specified in Appendix* (30)*).*

Theorem 4.1 demonstrates that taking the output policy $\widehat{\pi}$ as a random one selected from $\{\pi_{w_t}\}_{1 \leq i \leq T}$ following some distribution, the proposed ESPO algorithm achieves convergence to a globally

---

[2]Throughout this paper, the standard notation $\widetilde{O}(\cdot)$ indicates the order of a function with all constant terms hidden.

optimal policy $\pi^\star$ within the feasible safe set, following the convergence rate of $\widetilde{O}\left(\sqrt{\frac{SA}{(1-\gamma)^3 T}}\right)$. The convergence rate for constraint violations towards 0 is also $\widetilde{O}\left(\sqrt{\frac{SA}{(1-\gamma)^3 T}}\right)$. While note that the implementation of Algorithm 1 in practice only need to output the final $\widehat{\pi} = \pi_{w_T}$ for simplicity. The randomized procedure is only used for theoretical analysis.

We observe that ESPO enjoys the same convergence rate as the well-known primal safe RL algorithm — CRPO [53]. In addition, Theorem (4.1) directly indicates the same convergence rate guarantee for PCRPO [30] — the three-mode optimization framework that our ESPO refer to, which closes the gap between practice and theoretical guarantees for PCRPO [30]. Technically, to handle the variation in ESPO's update rules across a three-mode optimization process compared to CRPO, deriving the results necessitates to overcome additional challenges by tailoring a new distribution probability for the algorithm that is used to randomly select policies from $\{\pi_{w_t}\}_{1 \le i \le T}$.

Besides the efficient convergence, in the following, we present two advantages of ESPO in terms of both optimization benefits and sample efficiency; the proof are provided in Appendix A.4 and A.5 respectively.

**II: Efficient optimization with reduced oscillation.** Shown qualitatively in Figure 1, compared to other primal safe RL algorithms (such as CRPO), our proposed ESPO can significantly increase the ratios of iterations for maximizing the reward objective within the (relaxed) soft safe region by reducing oscillation across the safe region boundary. We provide a rigorous quantitative analysis for such advancement as below:

**Proposition 4.2.** *Suppose CRPO [53] and ESPO (ours) are initialized at an identical point $w_0 \in \mathbb{R}^{|\mathcal{S}||\mathcal{A}|}$. Denote the set of iterations that CRPO updates according to the reward objective as $\mathcal{B}_r^{\mathsf{CRPO}}$. Then by adaptively choosing the parameters $(x_t^r, x_t^c)$ of Algorithm 1, if there exist iteration $t_{\mathsf{in}} < T$ such that $t \in \mathcal{B}_r \cup \mathcal{B}_{\mathsf{soft}}$, one has*

$$\forall t_{\mathsf{in}} \le t \le T: \quad t \in \mathcal{B}_r \cup \mathcal{B}_{\mathsf{soft}}, \tag{9a}$$

$$|\mathcal{B}_r| + |\mathcal{B}_{\mathsf{soft}}| = T - t_{\mathsf{in}} \ge \mathcal{B}_r^{\mathsf{CRPO}}. \tag{9b}$$

In words, (9a) shows that as long as ESPO (cf. Algorithm 1) enters the safe region that the constraint is violated at most $h^+$, it will stay and always (at least partially) optimizes the reward objective without oscillation across the safe region boundary. In addition, (9b) indicates that the proposed ESPO enables more iterations to maximize the reward objective inside the safe region with comparison to CRPO, accelerating the optimization towards the global optimal policy. These two theoretical guarantees are further corroborated by the phenomena in practice (shown in Table 3): ESPO spends more iterations (99.4% steps) on optimizing the reward objective inside the safe region compared to CRPO (35.6% steps), while only a few on solely cost objective.

**III: Sample efficiency with sample size manipulation.** Besides the efficient optimization of ESPO, the following proposition presents the provable sample efficiency of ESPO.

**Proposition 4.3.** *Consider any $0 \le \varepsilon_1, \varepsilon_2 \le \frac{1}{1-\gamma}$. To meet the following goals of performance gaps*

$$V_r^{\pi^\star}(\rho) - \mathbb{E}[V_r^{\widehat{\pi}}(\rho)] \le \varepsilon_1, \ \mathbb{E}[V_c^{\widehat{\pi}}(\rho)] - V_c^{\pi^\star}(\rho) \le \varepsilon_2, \tag{10}$$

*ESPO (Algorithm 1) needs fewer number of samples than that without the sample manipulation in Section 4.2.*

The result demonstrates that, considering the accuracy level/constraint violation requirements, the sample manipulation module contributes to a more sample-efficient algorithm ESPO (Algorithm 1). Additionally, the *conflict* between reward and cost gradients emerges as an effective metric for determining sample size requirements.

## 5 Experiments and Evaluation

To evaluate the effectiveness of our algorithm, we compare it with two key paradigms in safe RL frameworks. The first paradigm is based on the primal framework, including PCRPO [30] and CRPO [53] as the representative baselines. The second paradigm includes methods that leverage

the primal-dual framework, with PCPO [56], CUP [54], and PPOLag [32] serving as representative methodologies. Our algorithm is developed within the primal framework, thereby highlighting the importance of comparing it against these paradigmatic safe RL algorithms to clearly demonstrate its performance. Experiments are conducted using both primal and primal-dual benchmarks. The *Omnisafe*[3] [32] benchmark is leveraged for primal-dual based methods, where representative techniques such as PCPO [56], CUP [54], and PPOLag [32] generally exhibit stronger performance compared to existing primal methods like CRPO [53], a finding discussed in [27]. Additionally, we use the *Safety-MuJoCo*[4] [30] benchmark for primal-based methods. This benchmark, developed in 2024, is relatively new and primarily supports primal-based methods due to the specific implementation efforts involved. The detailed experimental settings are provided in Appendix D. Furthermore, to thoroughly evaluate the effectiveness of our method, we conduct a series of ablation experiments regarding different cost limits and sample manipulation techniques. In particular, we provide performance update analysis in terms of constraint violations. These experiments are specifically designed to dissect and understand the impact of various factors integral to our approach.

## 5.1 Experiments of Comparison with Primal-Based Methods

We deploy our algorithm on the *Safety-MuJoCo* benchmark and carry out experiments compared with representative primal algorithms, PCRPO [30] and CRPO [53]. Specifically, we conduct experiments on a set of challenging tasks, namely, *SafetyReacher-v4*, *SafetyWalker-v4*, *SafetyHumanoidStandup-v4*.

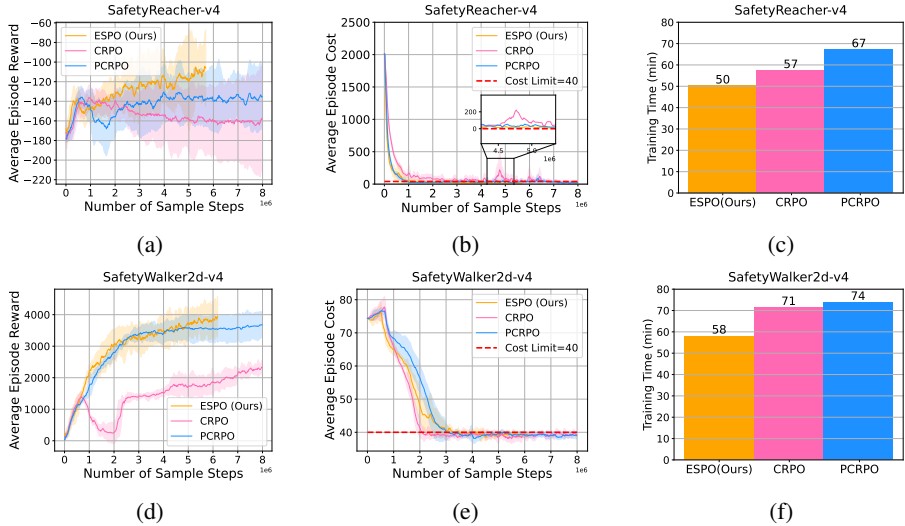

Figure 2: Compare our algorithm (ESPO) with PCRPO [30] and CRPO [53] on the *Safety-MuJoCo* benchmark. Our algorithm consistently and remarkably outperforms the SOTA baseline across multiple performance metrics, including reward maximization, safety assurance, and learning efficiency.

| Algorithm / Task | ESPO (Ours) | CRPO | PCRPO |
|---|---|---|---|
| *SafetyReacher-v4* | **5.7 M** | 8 M | 8 M |
| *SafetyWalker-v4* | **6.2 M** | 8 M | 8 M |
| *SafetyHumanoidStandup-v4* | **5.1 M** | 8 M | 8 M |

Table 1: Comparison of sampling steps with primal-based methods (The lower, the better). M denotes one million.

In the experiments conducted on the *SafetyReacher-v4* task, as depicted in Figures 2(a)-(c), our method demonstrates superior performance compared to SOTA primal baselines, CRPO and PCRPO. For instance, our method achieves better reward performance than CRPO and PCRPO. Another notable aspect of ESPO's performance is its training efficiency, which is largely attributed to sample manipulation. Specifically, as depicted in Table 1, while CRPO and PCRPO utilize 8 million

---

[3] https://github.com/PKU-Alignment/omnisafe
[4] https://github.com/SafeRL-Lab/Safety-MuJoCo

samples for the *SafetyReacher-v4* task, our method requires only 5.7 million samples for the same task. Crucially, our method improves reward and efficiency performance without sacrificing safety. However, CRPO and PCRPO are struggling to ensure safety during policy learning. Ensuring safety is a pivotal aspect of RL in safety-critical environments. The experiment results indicate that our method's ability to balance safety with other performance metrics is a significant improvement. As illustrated in Figures 2(d)-(f), our comparison experiments on the challenging *SafetyWalker-v4* task, yielding findings consistent with those observed in *SafetyReacher-v4* tasks. Due to space limits, additional experiments on *SafetyHumanoidStandup-v4* are postponed to Appendix D.

## 5.2 Experiments of Comparison with Primal-Dual-Based Methods

The *Omnisafe* Benchmark is a popular platform for evaluating the performance of safe RL algorithms. To further examine the effectiveness of our method, we have implemented our algorithm within the *Omnisafe* framework and conducted an extensive series of experiments compared with SOTA primal-dual-based baselines, e.g., PPOLag [32], CUP [54] and PCPO [56], focusing mainly on challenging tasks such as *SafetyHopperVelocity-v1* and *SafetyAntVelocity-v1*.

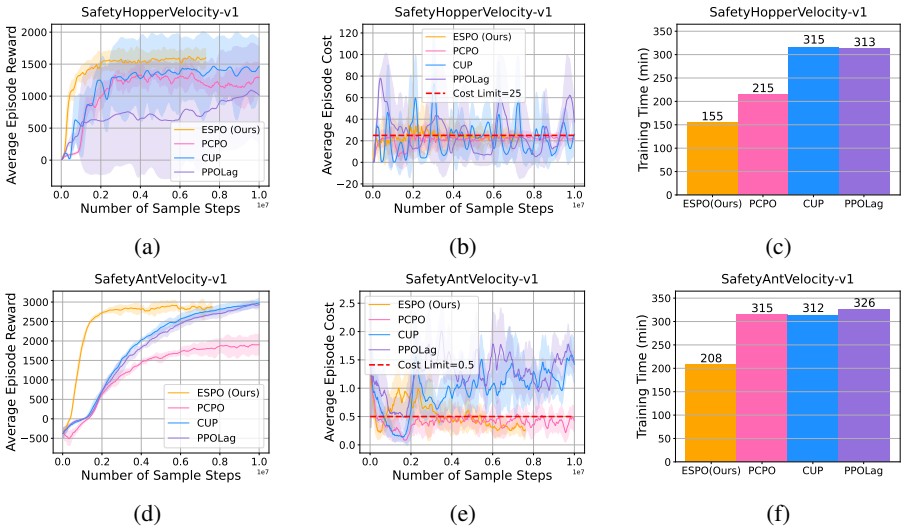

Figure 3: Compare our algorithm (ESPO) with PCPO [56], CUP [54] and PPOLag [32] on the *Omnisafe* benchmark. Our algorithm performs significantly better than the SOTA baselines regarding reward, safety, and efficiency performance.

| Algorithm
Task | ESPO (Ours) | PCPO | CUP | PPOLag |
|---|---|---|---|---|
| *SafetyHopperVelocity-v1* | **7.3 M** | 10 M | 10 M | 10 M |
| *SafetyAntVelocity-v1* | **7.6 M** | 10 M | 10 M | 10 M |

Table 2: Comparison of sampling steps with primal-dual based methods (The lower, the better). M denotes one million samples.

The efficacy of our algorithm, ESPO, is demonstrated in Figures 3(a)-(c), where it is benchmarked against SOTA baselines on the *SafetyHopperVelocity-v1* tasks. Firstly, ESPO is remarkably able to achieve better reward performance than the SOTA primal-dual-based baselines. Secondly, a critical aspect of our algorithm is its capability to ensure safety. It is particularly significant considering that some of the compared baselines, such as CUP [54] and PPOLag [32], struggle to maintain safety within the same task parameters. Thirdly, an outstanding feature of ESPO is its efficiency, as evidenced by approximately half the training time required compared to the SOTA baselines like CUP and PPOLag. This efficiency in training time demonstrates ESPO's practicality for use in various applications, especially where computational resources and time are constraints. Moreover, while PCPO [56] manages to ensure safety, its reward performance is inferior to ESPO's. PCPO also requires more training time than ESPO, underscoring our algorithm's reward, safety performance, and training efficiency advantages. Particularly, as illustrated in Table 2, across the entire training period, all the benchmark baselines, including PCPO, CUP, and PPOLag, utilized 10 million samples for tasks on *SafetyHopperVelocity-v1*. In contrast, our method required only 7.3 million samples for the *SafetyHopperVelocity-v1* task. The trends observed in the performance of our algorithm on the

*SafetyHopperVelocity-v1* task are similarly reflected in the results presented in Figures 3(d)-(f), about the *SafetyAntVelocity-v1* task. These findings further prove the effectiveness of ESPO in various tasks. Note that the reduction in samples may not equate to a corresponding reduction in training time, as this can vary depending on the characteristics of the benchmarks and the algorithms applied to different tasks. Factors such as the action space of the task and the settings of parallel processing supported by the benchmark can influence the overall training time.

These results on *Omnisafe* tasks further highlight the strengths of ESPO in improving reward performance with safety assurance while maintaining greater efficiency in training. The ability of ESPO validates its potential as an effective solution for further exploration and application in real-world environments.

### 5.3 Ablation Experiments

We conducted ablation studies focusing on various cost limits, sample sizes, learning rates, gradient weights, and update styles to further assess our method's effectiveness. These studies are crucial for gaining deeper insights into our method, highlighting its strengths, and identifying potential areas for improvement. Through this evaluation, we aim to demonstrate the adaptability of our method, confirming its applicability and efficacy across a broad spectrum of safe RL scenarios. Due to space limits, details of the ablation studies are provided in Appendix C.

## 6 Conclusion

In the study, we improved the efficiency of safe RL through a three-mode optimization scheme employing sample manipulation. We provide an in-depth theoretical analysis of convergence, stability, and sample complexity. These theoretical insights inform a practical algorithm for safety-critical control. Extensive experiments on two major benchmarks, *Safety-MuJoCo* and *Omnisafe*, indicate that our method not only surpasses the SOTA baselines in terms of efficiency but also achieves higher reward performance while maintaining safety. Moving forward, we plan to assess our method's capabilities in real world control applications to further expand its influential reach into safety-critical domains. Impact and limitation statements are provided in Appendix E.

### Acknowledgement

The work of L. Shi is supported in part by the Resnick Institute and Computing, Data, and Society Postdoctoral Fellowship at California Institute of Technology. The work of A. Wierman is supported in part from CNS-2146814, CPS-2136197, CNS-2106403, and NGSDI-2105648. M. Jin acknowledges the support from NSF ECCS-2331775. The work of S. Gu is supported by funds from the Prof. Spanos' Andrew S. Grove Endowed Chair.

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

# Appendix

## A  Proof of the theoretical analysis

Inspired by [53], the theoretical results in this section are established by tailoring to our algorithm ESPO to ensure the key recursion relation still hold for the proposed complex update rules — different update rules in three different modes.

### A.1  Preliminaries

To proceed, we first introduce some notations and invoke several key facts and results that have been derived by prior arts.

**Notation.**  We recall and introduce some useful notation throughout this section.

- $\bar{Q}_t^r, \bar{Q}_t^c$: this two function represent the policy evaluation results from Algoriathm 1, namely, the estimates of true Q-functions $Q_r^{w_t}, Q_c^{w_t}$.

- $\eta$: the learning rate of the NPG update rule in Algoriathm 1.

- $\mathcal{B}_{\text{soft}}^{\text{no}}, \mathcal{B}_{\text{soft}}^{\text{conf}}$: we denote the set of iterations when Algorithm 1 executes (4) (resp. (3)) as $\mathcal{B}_{\text{soft}}^{\text{no}}$ (resp. $\mathcal{B}_{\text{soft}}^{\text{conf}}$).

- $(x_t^r, x_t^c)$: when the iteration $t \in \mathcal{B}_{\text{soft}}^{\text{no}}$ (no conflict between the gradients of reward and cost objectives), $x_t^r$ (resp. $x_t^c$) represents the weight of the gradient w.r.t. the reward objective (resp. the cost function). So it is easily verified that $0 \le x_t^r, x_t^c \le 1$ and $x_t^r + x_t^c = 1$.

- $(y_t^r, y_t^c)$: when the iteration $t \in \mathcal{B}_{\text{soft}}^{\text{conf}}$ (the gradients of reward and cost objectives are conflict with each other), $y_t^r$ (resp. $y_t^c$) represents the weight of the gradient w.r.t. the reward objective (resp. the cost function). So it is easily verified that $y_t^r, y_t^c \ge 0$.

- $v_{\max}$: without loss of generality, we assume $r(s, a) \in [0, v_{\max}]$ and $c_i(s, a) \in [0, v_{\max}]$ for all $1 \le i \le n$.

- $h^+, h^-$: for simplicity, we let $h_t^+ = h^+, h_t^- = h^-$ for all $1 \le t \le T$.

**Lemma A.1** (Performance difference lemma [34] ). *For any policies $\pi$, $\pi'$ and initial distribution $\rho$, one has*

$$\forall i \in \{c, r\}: \quad V_i^\pi(\rho) - V_i^{\pi'}(\rho) = \frac{1}{1 - \gamma} \mathbb{E}_{s \sim d_\rho} \left[ \mathbb{E}_{a \sim \pi(\cdot|s)} [A_i^{\pi'}(s, a)] \right], \tag{11}$$

*where $V_i^\pi(\rho)$ and $d_\rho$ denote the accumulated reward (cost) function and state-action visitation distribution under policy $\pi$ when the initial state distribution is $\rho$. Here, $A_i^{\pi'}(s, a) = Q_i^{\pi'}(s, a) - V_i^{\pi'}(s)$ is the advantage function of policy $\pi$ over state-action pair $(s, a)$.*

**Lemma A.2.** *Considering the approximated NPG update rule and Algorithm 1 in the tabular setting, the NPG update in four possible diverse modes take the form:*

$$\begin{cases} w_{t+1} = w_t + \frac{\eta}{1-\gamma} \bar{Q}_t^r \quad and \quad \pi_{w_{t+1}}(a|s) = \pi_{w_t}(a|s) \frac{\exp\left(\frac{\eta \bar{Q}_t^r(s,a)}{(1-\gamma)}\right)}{Z_t^r(s)}, & \text{if } t \in \mathcal{B}_r \\[3mm] w_{t+1} = w_t + \frac{\eta\left(x_t^r \bar{Q}_t^r + x_t^c \bar{Q}_t^c\right)}{1-\gamma} \quad and \quad \pi_{w_{t+1}}(a|s) = \pi_{w_t}(a|s) \frac{\exp\left(\frac{\eta(x_t^r \bar{Q}_t^r(s,a) + x_t^c \bar{Q}_t^c(s,a))}{(1-\gamma)}\right)}{Z_t^{r,c,1}(s)}, & \text{if } t \in \mathcal{B}_{\text{soft}}^{\text{no}} \\[3mm] w_{t+1} = w_t + \frac{\eta\left(y_t^r \bar{Q}_t^r + y_t^c \bar{Q}_t^c\right)}{1-\gamma} \quad and \quad \pi_{w_{t+1}}(a|s) = \pi_{w_t}(a|s) \frac{\exp\left((\frac{\eta(y_t^r \bar{Q}_t^r(s,a) + y_t^c \bar{Q}_t^c(s,a))}{(1-\gamma)}\right)}{Z_t^{r,c,2}(s)}, & \text{if } t \in \mathcal{B}_{\text{soft}}^{\text{conf}} \\[3mm] w_{t+1} = w_t + \frac{\eta}{1-\gamma} \bar{Q}_t^c \quad and \quad \pi_{w_{t+1}}(a|s) = \pi_{w_t}(a|s) \frac{\exp\left(\frac{\eta \bar{Q}_t^c(s,a)}{(1-\gamma)}\right)}{Z_t^c(s)}, & \text{if } t \in \mathcal{B}_c \end{cases} \tag{12}$$

*where*

$$Z_t^r(s) = \sum_{a \in \mathcal{A}} \pi_{w_t}(a|s) \exp\left(\frac{\eta \bar{Q}_t^r(s,a)}{1-\gamma}\right),$$

$$Z_t^{r,c,1}(s) = \sum_{a \in \mathcal{A}} \pi_{w_t}(a|s) \exp\left(\frac{\eta \left(x_t^r \bar{Q}_t^r(s,a) + x_t^c \bar{Q}_t^c(s,a)\right)}{(1-\gamma)}\right)$$

$$Z_t^c(s) = \sum_{a \in \mathcal{A}} \pi_{w_t}(a|s) \exp\left(\frac{\eta \bar{Q}_t^c(s,a)}{1-\gamma}\right),$$

$$Z_t^{r,c,2}(s) = \sum_{a \in \mathcal{A}} \pi_{w_t}(a|s) \exp\left(\frac{\eta \left(y_t^r \bar{Q}_t^r(s,a) + y_t^c \bar{Q}_t^c(s,a)\right)}{(1-\gamma)}\right). \tag{13}$$

*Proof.* The first line of (12) has been verified by [Lemma 5.6. [2]]. Following the same proof pipeline for the update rules of Algorithm 1 in different modes completes the proof. □

## A.2 Key lemmas

The proof of Theorem 4.1 heavily count on several key lemmas in the following.

First, we introduce the performance improvement bound for the update rules of Algorithm 1 in different modes, which is a fundamental result for its convergence; the proof is postponed to Appendix A.6.1.

**Lemma A.3** (Performance improvement bound for approximated NPG). *Consider any initial state distribution $\rho$ and the iterate $\pi_{w_t}$ generated by Algorithm 1 at time step $t$. One has when iteration $t \in \mathcal{B}_r$:*

$$V_r^{\pi_{w_{t+1}}}(\rho) - V_r^{\pi_{w_t}}(\rho) \tag{14}$$

$$\geq \frac{1-\gamma}{\eta} \mathbb{E}_{s \sim \rho} \left(\log Z_t^r(s) - \frac{\eta}{1-\gamma} V_r^{\pi_{w_t}}(s) + \frac{\eta}{1-\gamma} \sum_{a \in \mathcal{A}} \pi_{w_t}(a|s) \left|\bar{Q}_t^r(s,a) - Q_r^{\pi_{w_t}}(s,a)\right|\right)$$

$$- \frac{1}{1-\gamma} \mathbb{E}_{s \sim d_\rho} \sum_{a \in \mathcal{A}} \pi_{w_t}(a|s) \left|\bar{Q}_t^r(s,a) - Q_r^{\pi_{w_t}}(s,a)\right|$$

$$- \frac{1}{1-\gamma} \mathbb{E}_{s \sim d_\rho} \sum_{a \in \mathcal{A}} \pi_{w_{t+1}}(a|s) \left|\bar{Q}_t^r(s,a) - Q_r^{\pi_{w_t}}(s,a)\right| := \mathsf{diff}_t^r. \tag{15}$$

*Similarly, we have*

$$\forall t \in \mathcal{B}_c : \quad V_c^{\pi_{w_{t+1}}}(\rho) - V_c^{\pi_{w_t}}(\rho) \geq \mathsf{diff}_t^c, \tag{16}$$

*and then*

$$\begin{cases} x_t^r \left(V_r^{\pi_{w_{t+1}}}(\rho) - V_r^{\pi_{w_t}}(\rho)\right) + x_t^c \left(V_c^{\pi_{w_{t+1}}}(\rho) - V_c^{\pi_{w_t}}(\rho)\right) \geq x_t^r \mathsf{diff}_t^r + x_t^c \mathsf{diff}_t^c & \text{if } t \in \mathcal{B}_{\mathsf{soft}}^{\mathsf{no}} \\ y_t^r \left(V_r^{\pi_{w_{t+1}}}(\rho) - V_r^{\pi_{w_t}}(\rho)\right) + y_t^c \left(V_c^{\pi_{w_{t+1}}}(\rho) - V_c^{\pi_{w_t}}(\rho)\right) \geq y_t^r \mathsf{diff}_t^r + y_t^c \mathsf{diff}_t^c & \text{if } t \in \mathcal{B}_{\mathsf{soft}}^{\mathsf{conf}}. \end{cases} \tag{17}$$

Armed with above lemma, now we can control the performance gap between the current poliy $\pi_{w_t}$ and the optimal policy $\pi^\star$ in the following lemma; the proof is postponed to Appendix A.6.2.

**Lemma A.4** (Suboptimality gap bound for update rules of Algorithm 1). *Consider the approximated NPG updates in (12). When iteration $t \in \mathcal{B}_r$, denoting the visitation distribution under the optimal policy as $d^\star$, we have*

$$V_r^{\pi^*}(\rho) - V_r^{\pi_{w_t}}(\rho)$$

$$\leq \frac{1}{\eta} \mathbb{E}_{s \sim d^*} (D_{KL}(\pi^*||\pi_{w_t}) - D_{KL}(\pi^*||\pi_{w_{t+1}})) + \frac{2\eta|\mathcal{S}||\mathcal{A}|v_{\max}^2}{(1-\gamma)^3} + \frac{3(1+\eta v_{\max})}{(1-\gamma)^2} \|Q_{\pi_{w_t}}^r - \bar{Q}_r^{\pi_{w_t}}\|_2$$

$$:= \mathsf{gap}_t^r \tag{18}$$

*Similarly, we have*

$$\forall t \in \mathcal{B}_c : \quad V_c^{\pi^*}(\rho) - V_c^{\pi_{w_t}}(\rho) \leq \mathsf{gap}_t^c. \tag{19}$$

*In addition, for other iterations: if $t \in \mathcal{B}_{\mathsf{soft}}^{\mathsf{no}}$, we have*

$$x_t^r \left( V_r^{\pi^*}(\rho) - V_r^{\pi_{w_t}}(\rho) \right) + x_t^c \left( V_c^{\pi^*}(\rho) - V_c^{\pi_{w_t}}(\rho) \right)$$

$$\leq \frac{1}{\eta} \mathbb{E}_{s \sim d^*} (D_{KL}(\pi^* || \pi_{w_t}) - D_{KL}(\pi^* || \pi_{w_{t+1}})) + \frac{2\eta v_{\max}^2 |\mathcal{S}||\mathcal{A}|}{(1-\gamma)^3}$$

$$+ \frac{3(1 + \eta v_{\max})}{(1-\gamma)^2} \left[ x_t^r \left\| Q_r^{\pi_{w_t}}(s,a) - \bar{Q}_t^i(s,a) \right\|_2 + x_t^c \left\| Q_c^{\pi_{w_t}}(s,a) - \bar{Q}_t^c(s,a) \right\|_2 \right], \tag{20}$$

*and if $t \in \mathcal{B}_{\mathsf{soft}}^{\mathsf{conf}}$, we have*

$$y_t^r \left( V_r^{\pi^*}(\rho) - V_r^{\pi_{w_t}}(\rho) \right) + y_t^c \left( V_c^{\pi^*}(\rho) - V_c^{\pi_{w_t}}(\rho) \right)$$

$$\leq \frac{1}{\eta} \mathbb{E}_{s \sim d^*} (D_{KL}(\pi^* || \pi_{w_t}) - D_{KL}(\pi^* || \pi_{w_{t+1}})) + \frac{2\eta v_{\max}^2 (y_t^r + y_t^c)|\mathcal{S}||\mathcal{A}|}{(1-\gamma)^3}$$

$$+ \frac{3(1 + \eta v_{\max})}{(1-\gamma)^2} \left[ x_t^r \left\| Q_r^{\pi_{w_t}}(s,a) - \bar{Q}_t^i(s,a) \right\|_2 + x_t^c \left\| Q_c^{\pi_{w_t}}(s,a) - \bar{Q}_t^c(s,a) \right\|_2 \right], \tag{21}$$

Now we are ready to develop a key lemma that is associated with the expectation of the performance gap directly. The proof is provided in Appendix A.6.3

**Lemma A.5.** *In the tabular setting, consider any $0 < \delta < 1$ and suppose the iterations of policy evaluation obey $T_{\mathsf{pi}} = \widetilde{O}\big(\frac{T \log(\frac{|\mathcal{S}||\mathcal{A}|}{\delta})}{(1-\gamma)^3 |\mathcal{S}||\mathcal{A}|}\big)$. With probability at least $1 - \delta$, applying Algorithm 1 leads to*

$$\eta \sum_{t \in \mathcal{B}_r} (V_r^{\pi^*}(\rho) - V_r^{\pi_{w_t}}(\rho)) + \eta \sum_{t \in \mathcal{B}_{\mathsf{soft}}^{\mathsf{no}}} x_r^t (V_r^{\pi_{w_t}}(\rho) - V_r^{\pi^*}(\rho)) + \eta \sum_{t \in \mathcal{B}_{\mathsf{soft}}^{\mathsf{conf}}} y_r^t (V_r^{\pi_{w_t}}(\rho) - V_r^{\pi^*}(\rho))$$

$$+ \eta h^+ |\mathcal{B}_c| - \eta h^- \sum_{t \in \mathcal{B}_{\mathsf{soft}}^{\mathsf{no}}} x_r^t - \eta h^- \sum_{t \in \mathcal{B}_{\mathsf{soft}}^{\mathsf{conf}}} y_r^t$$

$$\leq \mathbb{E}_{s \sim d^*} D_{KL}(\pi^* || \pi_{w_0}) + \frac{2\eta^2 v_{\max}^2 |\mathcal{S}||\mathcal{A}|}{(1-\gamma)^3} \left[ (T - |\mathcal{B}_{\mathsf{soft}}^{\mathsf{conf}}|) + \sum_{t \in \mathcal{B}_{\mathsf{soft}}^{\mathsf{conf}}} (y_t^c + y_t^r) \right] + \frac{3\eta(1 + \eta v_{\max})}{(1-\gamma)^2} \epsilon_{\mathsf{pi}}$$

$$\tag{22}$$

$$\leq \mathbb{E}_{s \sim d^*} D_{KL}(\pi^* || \pi_{w_0}) + \frac{4\eta^2 v_{\max}^2 |\mathcal{S}||\mathcal{A}|T}{(1-\gamma)^3} + \frac{3\eta(1 + \eta v_{\max}) \sqrt{|\mathcal{S}||\mathcal{A}|T}}{(1-\gamma)^{1.5}}, \tag{23}$$

*where*

$$\epsilon_{\mathsf{pi}} := \sum_{t \in \mathcal{B}_r} \left\| Q_r^{\pi_{w_t}} - \bar{Q}_t^r \right\|_2 + \sum_{t \in \mathcal{B}_c} \left\| Q_c^{\pi_{w_t}} - \bar{Q}_t^c \right\|_2 +$$

$$+ \sum_{t \in \mathcal{B}_{\mathsf{soft}}^{\mathsf{no}}} \left( x_t^r \| Q_r^{\pi_{w_t}} - \bar{Q}_t^r \|_2 + x_t^c \| Q_c^{\pi_{w_t}} - \bar{Q}_t^c \|_2 \right)$$

$$+ \sum_{t \in \mathcal{B}_{\mathsf{soft}}^{\mathsf{conf}}} \left( y_t^r \| Q_r^{\pi_{w_t}} - \bar{Q}_t^r \|_2 + y_t^c \| Q_c^{\pi_{w_t}} - \bar{Q}_t^c \|_2 \right). \tag{24}$$

Finally, we introduce the following lemma which indicates the number of iterations that optimize the reward objective is in the order of $T$ as long as $h^+$ and $h^-$ are chosen properly. The proof is provided in Appendix A.6.4

**Lemma A.6** (The frequency of optimizing reward objective). *Consider any $0 < \delta < 1$ and $h^- = 0$. Suppose*

$$\frac{1}{2}\eta h^+ T \geq \mathbb{E}_{s \sim d^*} D_{KL}(\pi^* || \pi_{w_0}) + \frac{4\alpha^2 v_{\max}^2 |\mathcal{S}||\mathcal{A}|T}{(1-\gamma)^3} + \frac{3\eta(1 + \eta v_{\max}) \sqrt{|\mathcal{S}||\mathcal{A}|T}}{(1-\gamma)^{1.5}}, \tag{25}$$

*then with probability at least $1 - \delta$, the following fact holds*

1. $\mathcal{B}_r \cup \mathcal{B}_{\text{soft}} \neq \emptyset$.

2. *Either of the following claims holds:*
   *(a) $|\mathcal{B}_r \cup \mathcal{B}_{\text{soft}}| \geq T/2$;*
   *(b) The weighted performance gap is non-positive:*

$$\sum_{t \in \mathcal{B}_r} (V_r^{\pi^*}(\rho) - V_r^{\pi_{w_t}}(\rho)) + \sum_{t \in \mathcal{B}_{\text{soft}}^{\text{no}}} x_r^t (V_r^{\pi^*}(\rho) - V_r^{\pi_{w_t}}(\rho))$$

$$+ \sum_{t \in \mathcal{B}_{\text{soft}}^{\text{conf}}} y_r^t (V_r^{\pi^*}(\rho) - V_r^{\pi_{w_t}}(\rho)) \leq 0. \tag{26}$$

## A.3  Proof of Theorem 4.1

Now we are ready to provide the proof for Theorem 4.1.

Recall the goal is to prove

$$V_r^{\pi^*}(\rho) - \mathbb{E}[V_r^{\widehat{\pi}}(\rho)] \leq \widetilde{O}\left(\sqrt{\frac{SA}{(1-\gamma)^3 T}}\right), \tag{27}$$

where the expectation is taken with respect to a weighted average over all $\{\pi_{w_t}\}_{1 \leq t \leq T}$.

We still consider the modes when the policy evaluation results are accurate such that

$$\epsilon_{\text{pi}} \leq \sqrt{(1-\gamma)|\mathcal{S}||\mathcal{A}|T}, \tag{28}$$

which combined with Lemma A.5 yields

$$\eta \sum_{t \in \mathcal{B}_r} (V_r^{\pi^*}(\rho) - V_r^{\pi_{w_t}}(\rho)) + \eta \sum_{t \in \mathcal{B}_{\text{soft}}^{\text{no}}} \left[ x_r^t (V_r^{\pi^*}(\rho) - V_r^{\pi_{w_t}}(\rho)) + x_c^t (V_c^{\pi^*}(\rho) - V_c^{\pi_{w_t}}(\rho)) \right]$$

$$+ \eta \sum_{t \in \mathcal{B}_{\text{soft}}^{\text{conf}}} \left[ y_r^t (V_r^{\pi^*}(\rho) - V_r^{\pi_{w_t}}(\rho)) + y_c^t (V_c^{\pi^*}(\rho) - V_c^{\pi_{w_t}}(\rho)) \right]$$

$$+ \eta h^+ |\mathcal{B}_c| - \eta h^- \sum_{t \in \mathcal{B}_{\text{soft}}^{\text{no}}} x_r^t - \eta h^- \sum_{t \in \mathcal{B}_{\text{soft}}^{\text{conf}}} y_r^t$$

$$\leq \eta \mathbb{E}_{s \sim d^*} D_{\text{KL}}(\pi^* || \pi_{w_0}) + \frac{2\eta^2 v_{\max}^2 |\mathcal{S}||\mathcal{A}|T}{(1-\gamma)^3} + \frac{3\eta(1 + \eta v_{\max})\sqrt{|\mathcal{S}||\mathcal{A}|T}}{(1-\gamma)^{1.5}}. \tag{29}$$

**The probability distribution associated with the expectation.**   Here, we let the weighs (probability distribution) to be proportion to

$$\begin{cases} 1 & \text{if } t \in \mathcal{B}_r \\ x_t^r & \text{if } t \in \mathcal{B}_{\text{soft}}^{\text{no}} \\ y_t^r & \text{if } t \in \mathcal{B}_{\text{soft}}^{\text{conf}} \\ 0 & \text{if } t \in \mathcal{B}_c, \end{cases} \tag{30}$$

which will be normalized by

$$T_{\text{weighted}}^r = |\mathcal{B}_r| + \sum_{t \in \mathcal{B}_{\text{soft}}^{\text{no}}} x_t^r + \sum_{t \in \mathcal{B}_{\text{soft}}^{\text{conf}}} y_t^r. \tag{31}$$

Then we introduce an important fact for $y_t^r$ and $y_t^c$. Recall that when $t \in \mathcal{B}_{\text{soft}}^{\text{conf}}$, keeping the weights $x_t^r$ and $x_t^c$ as the same as the mode $t \in \mathcal{B}_{\text{soft}}^{\text{no}}$ for the reward and cost, the gradient is constructed as

$$\mathbf{g}_t = x_t^r \left( \mathbf{g}_r - \frac{\mathbf{g}_r \cdot \mathbf{g}_c}{\|\mathbf{g}_c\|^2} \mathbf{g}_c \right) + x_t^c \left( \mathbf{g}_c - \frac{\mathbf{g}_c \cdot \mathbf{g}_r}{\|\mathbf{g}_r\|^2} \mathbf{g}_r \right)$$

$$= x_t^r \left( 1 + \frac{\cos\theta_{rc}^t \|\mathbf{g}_c\|}{\|\mathbf{g}_r\|} \right) \mathbf{g}_r + x_t^c \left( 1 + \frac{\cos\theta_{rc}^t \|\mathbf{g}_r\|}{\|\mathbf{g}_c\|} \right) \mathbf{g}_c, \tag{32}$$

which indicates

$$\forall t \in \mathcal{B}_{\mathsf{soft}}^{\mathsf{conf}}: \quad y_t^r = x_t^r \left(1 + \frac{\cos\theta_{rc}^t \|\mathbf{g}_c\|}{\|\mathbf{g}_r\|}\right) \geq x_t^r \quad \text{and} \quad y_t^c = x_t^c \left(1 + \frac{\cos\theta_{rc}^t \|\mathbf{g}_r\|}{\|\mathbf{g}_c\|}\right) \geq x_t^c, \tag{33}$$

since $\cos\theta_{rc}^t \geq 0$ as $t \in \mathcal{B}_{\mathsf{soft}}^{\mathsf{conf}}$. The above fact directly gives that letting $x_t^r \geq 1/2$

$$\text{If } |\mathcal{B}_{\mathsf{r}} \cup \mathcal{B}_{\mathsf{soft}}| \geq \frac{T}{2} : T_{\mathsf{weighted}}^r \geq \frac{T}{4}. \tag{34}$$

**The reward objective.** We first consider the performance gap w.r.t. the reward. Armed with above facts, we can see if (26) holds, with the weights in (30) then we directly have

$$V_r^{\pi^\star}(\rho) - \mathbb{E}[V_r^{\widehat{\pi}}(\rho)] \leq 0. \tag{35}$$

Otherwise, applying Lemma A.6 gives

$$\begin{aligned}
T_{\mathsf{weighted}}^r \eta &\left(V_r^{\pi^\star}(\rho) - \mathbb{E}[V_r^{\widehat{\pi}}(\rho)]\right) \\
&= \eta \sum_{t \in \mathcal{B}_{\mathsf{r}}} (V_r^{\pi^\star}(\rho) - V_r^{\pi_{w_t}}(\rho)) + \eta \sum_{t \in \mathcal{B}_{\mathsf{soft}}^{\mathsf{no}}} x_r^t (V_r^{\pi^\star}(\rho) - V_r^{\pi_{w_t}}(\rho)) \\
&\quad + \eta \sum_{t \in \mathcal{B}_{\mathsf{soft}}^{\mathsf{conf}}} y_r^t (V_r^{\pi^\star}(\rho) - V_r^{\pi_{w_t}}(\rho)) \\
&\leq \mathbb{E}_{s \sim d^*} D_{\mathsf{KL}}(\pi^*\|\pi_{w_0}) + \frac{2\eta^2 v_{\max}^2 |\mathcal{S}||\mathcal{A}|T}{(1-\gamma)^3} + \frac{3\eta(1 + \eta v_{\max})\sqrt{|\mathcal{S}||\mathcal{A}|T}}{(1-\gamma)^{1.5}},
\end{aligned} \tag{36}$$

which indicates

$$V_r^{\pi^\star}(\rho) - \mathbb{E}[V_r^{\widehat{\pi}}(\rho)] \leq \frac{2\sqrt{|\mathcal{S}||\mathcal{A}|}}{(1-\gamma)^{1.5}\sqrt{T}} \left(\mathbb{E}_{s \sim d^*} D_{\mathsf{KL}}(\pi^*\|\pi_{w_0}) + 4v_{\max}^2 + 6v_{\max}\right). \tag{37}$$

Here, the last inequality hols by letting the learning rate $\eta = (1-\gamma)^{1.5}/\sqrt{|\mathcal{S}||\mathcal{A}|T}$.

**Constraint violation.** Now we move on to the cost objective. Taking the probability distribution of the expectation in (30) as well, we have

$$\begin{aligned}
\mathbb{E}[&V_c^{\widehat{\pi}}(\rho)] - b \\
&\leq \frac{1}{T_{\mathsf{weighted}}^r} \left(\sum_{t \in \mathcal{B}_{\mathsf{r}}} V_c^{\pi_{w_t}}(\rho) + \sum_{t \in \mathcal{B}_{\mathsf{soft}}^{\mathsf{no}}} x_r^t V_c^{\pi_{w_t}}(\rho) + \sum_{t \in \mathcal{B}_{\mathsf{soft}}^{\mathsf{conf}}} y_r^t V_c^{\pi_{w_t}}(\rho)\right) - b \\
&\leq \frac{1}{T_{\mathsf{weighted}}^r} \left(\sum_{t \in \mathcal{B}_{\mathsf{r}}} \left(\overline{V}_c^{\pi_{w_t}}(\rho) - b\right) + \sum_{t \in \mathcal{B}_{\mathsf{soft}}^{\mathsf{no}}} x_r^t \left(\overline{V}_c^{\pi_{w_t}}(\rho) - b\right) + \sum_{t \in \mathcal{B}_{\mathsf{soft}}^{\mathsf{conf}}} y_r^t \left(\overline{V}_c^{\pi_{w_t}}(\rho) - b\right)\right) \\
&\quad + \frac{1}{T_{\mathsf{weighted}}^r} \left(\sum_{t \in \mathcal{B}_{\mathsf{r}}} \left|\overline{V}_c^{\pi_{w_t}}(\rho) - V_c^{\pi_{w_t}}(\rho)\right| + \sum_{t \in \mathcal{B}_{\mathsf{soft}}^{\mathsf{no}}} x_r^t \left|\overline{V}_c^{\pi_{w_t}}(\rho) - V_c^{\pi_{w_t}}(\rho)\right| \right. \\
&\qquad \left. + \sum_{t \in \mathcal{B}_{\mathsf{soft}}^{\mathsf{conf}}} y_r^t \left|\overline{V}_c^{\pi_{w_t}}(\rho) - V_c^{\pi_{w_t}}(\rho)\right|\right) \\
&\leq h^+ + \frac{1}{T_{\mathsf{weighted}}^r} \left(\sum_{t \in \mathcal{B}_{\mathsf{r}}} \left|Q_c^{\pi_{w_t}} - \overline{Q}_t^c\right| + \sum_{t \in \mathcal{B}_{\mathsf{soft}}^{\mathsf{no}}} x_r^t \left|Q_c^{\pi_{w_t}} - \overline{Q}_t^c\right| + \sum_{t \in \mathcal{B}_{\mathsf{soft}}^{\mathsf{conf}}} y_r^t \left|Q_c^{\pi_{w_t}} - \overline{Q}_t^c\right|\right) \\
&\leq h^+ + \frac{4}{T} \left(\sum_{t \in \mathcal{B}_{\mathsf{r}}} \left|Q_c^{\pi_{w_t}} - \overline{Q}_t^c\right| + \sum_{t \in \mathcal{B}_{\mathsf{soft}}^{\mathsf{no}}} x_r^t \left|Q_c^{\pi_{w_t}} - \overline{Q}_t^c\right| + \sum_{t \in \mathcal{B}_{\mathsf{soft}}^{\mathsf{conf}}} y_r^t \left|Q_c^{\pi_{w_t}} - \overline{Q}_t^c\right|\right). \tag{38}
\end{aligned}$$

where the last inequality holds by (34). Finally, also considering the mode when the policy evaluation error in (28), we have

$$\left( \sum_{t \in \mathcal{B}_r} \left| Q_c^{\pi_{w_t}} - \overline{Q}_t^c \right| + \sum_{t \in \mathcal{B}_{\mathrm{soft}}^{\mathrm{no}}} x_r^t \left| Q_c^{\pi_{w_t}} - \overline{Q}_t^c \right| + \sum_{t \in \mathcal{B}_{\mathrm{soft}}^{\mathrm{conf}}} y_r^t \left| Q_c^{\pi_{w_t}} - \overline{Q}_t^c \right| \right) \le \epsilon_{\mathrm{pi}} \le \sqrt{(1-\gamma)|\mathcal{S}||\mathcal{A}|T}.$$
(39)

Then without loss of generality, taking the tolerance level $h^- = 0$ and

$$h^+ = \frac{2\sqrt{|\mathcal{S}||\mathcal{A}|}}{(1-\gamma)^{1.5}\sqrt{T}} \left( \mathbb{E}_{s \sim d^*} D_{\mathrm{KL}}(\pi^* || \pi_{w_0}) + 4v_{\max}^2 + 6v_{\max} \right)$$
(40)

complete the proof by showing

$$\mathbb{E}[V_c^{\widehat{\pi}}(\rho)] - b \le h^+ + \frac{4}{T} \left( \sum_{t \in \mathcal{B}_r} \left| Q_c^{\pi_{w_t}} - \overline{Q}_t^c \right| + \sum_{t \in \mathcal{B}_{\mathrm{soft}}^{\mathrm{no}}} x_r^t \left| Q_c^{\pi_{w_t}} - \overline{Q}_t^c \right| + \sum_{t \in \mathcal{B}_{\mathrm{soft}}^{\mathrm{conf}}} y_r^t \left| Q_c^{\pi_{w_t}} - \overline{Q}_t^c \right| \right)$$
(41)

$$\le \frac{2\sqrt{|\mathcal{S}||\mathcal{A}|}}{(1-\gamma)^{1.5}\sqrt{T}} \left( \mathbb{E}_{s \sim d^*} D_{\mathrm{KL}}(\pi^* || \pi_{w_0}) + 4v_{\max}^2 + 6v_{\max} \right) + \frac{4\sqrt{|\mathcal{S}||\mathcal{A}|}}{(1-\gamma)^{1.5}\sqrt{T}}.$$
(42)

### A.4 Proof of proposition 4.2

We consider the ideal mode when the number of iterations of policy evaluation $T_{\mathrm{pi}} \to \infty$ such that the ground truth cost function $V_c^{\pi_{w_t}} = \overline{V}_{t_{\mathrm{in}}}^c$.

First, we will focus on verifying the fact in (9a). Recall that there exists an iteration $t_{\mathrm{in}} < T$ such that $t_{\mathrm{in}} \in \mathcal{B}_r \cup \mathcal{B}_{\mathrm{soft}}$. So for the next step $t = t_{\mathrm{in}} + 1$, we consider two different modes separately.

- *When $t_{\mathrm{in}} \in \mathcal{B}_r$.* In this mode, we directly have

$$V_c^{\pi_{w_{t_{\mathrm{in}}}}}(\rho) = \overline{V}_{t_{\mathrm{in}}}^c \le b - h^-.$$
(43)

  Then we know that for the next step $t = t_{\mathrm{in}} + 1$,

$$V_c^{\pi_{w_t}}(\rho) \le V_c^{\pi_{w_{t_{\mathrm{in}}}}}(\rho) + \eta \| \nabla_w V_r^{\pi_{w_{t_{\mathrm{in}}}}}(\rho) \|_2 \le b - h^- + \frac{2v_{\max}\eta}{1-\gamma} \le b + h^+,$$
(44)

  where the penultimate inequality holds by the bound of the policy gradient established in [53, Lemma 5], and the last inequality holds by when the learning rate $\eta$ is small enough such that

$$\frac{2v_{\max}\eta}{1-\gamma} \le \frac{2\sqrt{|\mathcal{S}||\mathcal{A}|}}{(1-\gamma)^{1.5}\sqrt{T}} \left( \mathbb{E}_{s \sim d^*} D_{\mathrm{KL}}(\pi^* || \pi_{w_0}) + 4v_{\max}^2 + 6v_{\max} \right) = h^+.$$
(45)

  The observation in (44) shows that the next time step $t = t_{\mathrm{in}} + 1 \in \mathcal{B}_r \cup \mathcal{B}_{\mathrm{soft}}$.

- When $t_{\mathrm{in}} \in \mathcal{B}_{\mathrm{soft}}$. One has

$$V_c^{\pi_{w_{t_{\mathrm{in}}}}}(\rho) = \overline{V}_{t_{\mathrm{in}}}^c \le b + h^+.$$
(46)

  Then we can adaptively choose the weights for the reward and cost function $x_t^c, x_t^r$. Invoking Lemma A.3, we have

$$\begin{cases} x_t^r \left( V_r^{\pi_{w_{t+1}}}(\rho) - V_r^{\pi_{w_t}}(\rho) \right) + x_t^c \left( V_c^{\pi_{w_t}}(\rho) - V_c^{\pi_{w_{t+1}}}(\rho) \right) \ge 0 & \text{if} \quad t \in \mathcal{B}_{\mathrm{soft}}^{\mathrm{no}} \\ y_t^r \left( V_r^{\pi_{w_{t+1}}}(\rho) - V_r^{\pi_{w_t}}(\rho) \right) + y_t^c \left( V_c^{\pi_{w_t}}(\rho) - V_c^{\pi_{w_{t+1}}}(\rho) \right) \ge 0 & \text{if} \quad t \in \mathcal{B}_{\mathrm{soft}}^{\mathrm{conf}}. \end{cases}$$
(47)

  Then observing that when $t = t_{\mathrm{in}}$, in the mode with $x_t^c = 1$, we have

$$\begin{cases} \left( V_c^{\pi_{w_{t_{\mathrm{in}}}}}(\rho) - V_c^{\pi_{w_t}}(\rho) \right) \ge 0 & \text{if} \quad t \in \mathcal{B}_{\mathrm{soft}}^{\mathrm{no}} \\ \left( V_c^{\pi_{w_{t_{\mathrm{in}}}}}(\rho) - V_c^{\pi_{w_t}}(\rho) \right) \ge 0 & \text{if} \quad t \in \mathcal{B}_{\mathrm{soft}}^{\mathrm{conf}}, \end{cases}$$
(48)

which implies that

$$V_c^{\pi_{w_t}}(\rho) \le V_c^{\pi_{w_{t_{in}}}}(\rho) \le b + h^+. \tag{49}$$

So we have the next time step $t = t_{in} + 1 \in \mathcal{B}_r \cup \mathcal{B}_{soft}$.

This implies that as long as $x_t^c, x_t^r$ are chosen properly ensuring (48) holds, we can achieve $t = t_{in} + 1 \in \mathcal{B}_r \cup \mathcal{B}_{soft}$.

Summing up the two modes and applying them recursively, we complete the proof of (9a).

Finally, to verify (9b), we suppose ESPO and CRPO are initialized at the same point. Then observing that ESPO and CRPO execute the same update rule until the iteration $t_{in} \in \mathcal{B}_r \cup \mathcal{B}_{soft}$. Then applying (9a), we know that

$$|\mathcal{B}_r \cup \mathcal{B}_{soft}| = T - t_{in}. \tag{50}$$

While CRPO may has some iterations later such that falls into $\mathcal{B}_c$. So we have the number of iterations when CRPO update according to the reward objective $\mathcal{B}_r^{CRPO} \le T - t_{in}$. We complete the proof.

### A.5 Proof of proposition 4.3

Recall the goal of the algorithm is to achieve

$$V_r^{\pi^\star}(\rho) - \mathbb{E}[V_r^{\widehat{\pi}}(\rho)] \le \varepsilon_1, \ \mathbb{E}[V_c^{\widehat{\pi}}(\rho)] - V_c^{\pi^\star}(\rho) \le \varepsilon_2. \tag{51}$$

with as few samples as possible.

We start from considering $V_r^{\pi^\star}(\rho) - \mathbb{E}[V_r^{\widehat{\pi}}(\rho)] \le \varepsilon_1$. We observe that if (26) holds, taking the expectation w.r.t. the probability distribution in (30), we directly have

$$V_r^{\pi^\star}(\rho) - \mathbb{E}[V_r^{\widehat{\pi}}(\rho)] \le 0 \le \varepsilon_1. \tag{52}$$

Otherwise, applying Lemma A.6 and (22) gives

$$V_r^{\pi^\star}(\rho) - \mathbb{E}[V_r^{\widehat{\pi}}(\rho)]$$
$$= \sum_{t \in \mathcal{B}_r}(V_r^{\pi^\star}(\rho) - V_r^{\pi_{w_t}}(\rho)) + \sum_{t \in \mathcal{B}_{soft}^{no}} x_r^t(V_r^{\pi^\star}(\rho) - V_r^{\pi_{w_t}}(\rho)) + \sum_{t \in \mathcal{B}_{soft}^{conf}} y_r^t(V_r^{\pi^\star}(\rho) - V_r^{\pi_{w_t}}(\rho))$$
$$\le \frac{1}{\eta}\mathbb{E}_{s \sim d^*}D_{KL}(\pi^* \| \pi_{w_0}) + \frac{2\eta v_{max}^2|\mathcal{S}||\mathcal{A}|T}{(1-\gamma)^3} + \frac{3(1+\eta v_{max})}{(1-\gamma)^2}\epsilon_{pi}. \tag{53}$$

The first two terms are independent to the sample size. So we focus on control $\frac{3(1+\eta v_{max})}{(1-\gamma)^2}\epsilon_{pi}$ to meet the goal, namely, we need to achieve

$$\epsilon_{pi} = \sum_{t \in \mathcal{B}_r}\left\|Q_r^{\pi_{w_t}} - \bar{Q}_t^r\right\|_2 + \sum_{t \in \mathcal{B}_{soft}^{no}}\left(x_t^r\|Q_r^{\pi_{w_t}} - \bar{Q}_t^r\|_2 + x_t^c\|Q_c^{\pi_{w_t}} - \bar{Q}_t^c\|_2\right)$$
$$+ \sum_{t \in \mathcal{B}_{soft}^{conf}}\left(y_t^r\|Q_r^{\pi_{w_t}} - \bar{Q}_t^r\|_2 + y_t^c\|Q_c^{\pi_{w_t}} - \bar{Q}_t^c\|_2\right) \le \varepsilon_1' \tag{54}$$

for some $\varepsilon_1' \le \varepsilon_1$.

To continue, without loss of generality, we let $x_t^r = 1$, $x_t^c = 0$, and $|\mathcal{B}_r| = 0$ (in this mode, the sampling approach is fixed), we have

$$\epsilon_{pi} = \sum_{t \in \mathcal{B}_{soft}^{no}}\left\|Q_r^{\pi_{w_t}} - \bar{Q}_t^r\right\|_2 + \sum_{t \in \mathcal{B}_{soft}^{conf}} y_t^r\|Q_r^{\pi_{w_t}} - \bar{Q}_t^r\|_2$$
$$= \sum_{t \in \mathcal{B}_{soft}^{no}}\left\|Q_r^{\pi_{w_t}} - \bar{Q}_t^r\right\|_2 + \sum_{t \in \mathcal{B}_{soft}^{conf}}\left(1 + \frac{\cos\theta_{rc}^t\|\mathbf{g}_c\|}{\|\mathbf{g}_r\|}\right)\|Q_r^{\pi_{w_t}} - \bar{Q}_t^r\|_2$$
$$= \sum_{t \in \mathcal{B}_{soft}^{no}} \delta_t + \sum_{t \in \mathcal{B}_{soft}^{conf}}\left(1 + \frac{\cos\theta_{rc}^t\|\mathbf{g}_c\|}{\|\mathbf{g}_r\|}\right)\delta_t \tag{55}$$

where the penultimate inequality holds by the relation between $y_t^r, x_t^r$ in (33), and the last inequality follows from denoting $\left\| Q_r^{\pi_{w_t}} - \bar{Q}_t^r \right\|_2 = \delta_t$.

Now we are ready to show the advantages of using different batch size for different modes when $t \in \mathcal{B}_{\text{soft}}^{\text{no}}$ or $t \in \mathcal{B}_{\text{soft}}^{\text{conf}}$. We make the following assumption about the relation between $\delta_t$ and sample size (the number of iterations for the policy evaluation of Algorithm 1), which is qualitatively consistent with the policy evaluation bound in [53, Lemma 2].

**Assumption A.7.** Suppose for any $t \in \mathcal{B}_{\text{soft}}$, when the sample size varies around some basic size, the possible feasible $\delta_t$ is in the range such that $\delta_t = Y - \alpha s_t^{\text{B}}$ such that $Y$ is some small constant and $s_t^{\text{B}}$ is the sample size used for policy evaluation at $t$-th iteration.

With the above assumption in hand, (55) can be written as

$$\epsilon_{\text{pi}} = \sum_{t \in \mathcal{B}_{\text{soft}}^{\text{no}}} Y - \alpha s_t^{\text{B}} + \sum_{t \in \mathcal{B}_{\text{soft}}^{\text{conf}}} \left( 1 + \frac{\cos \theta_{rc}^t \|\mathbf{g}_c\|}{\|\mathbf{g}_r\|} \right) (Y - \alpha s_t^{\text{B}}) = \varepsilon_1'. \tag{56}$$

If there is no adaptive sampling, then we have $s_t^{\text{B}} = s_{t'}^{\text{B}}$ for any $t, t' \in \mathcal{B}_{\text{soft}}$, which leads to the total number of samples as

$$N_{\text{all}} = s_{\text{batch}} |\mathcal{B}_{\text{soft}}| = \frac{Y |\mathcal{B}_{\text{soft}}|}{\alpha} - \frac{\widetilde{\varepsilon}_1 |\mathcal{B}_{\text{soft}}|}{\alpha \left( |\mathcal{B}_{\text{soft}}^{\text{no}}| + \sum_{t \in \mathcal{B}_{\text{soft}}^{\text{conf}}} \left( 1 + \frac{\cos \theta_{rc}^t \|\mathbf{g}_c\|}{\|\mathbf{g}_r\|} \right) \right)}, \tag{57}$$

where $s_{\text{batch}}$ is the number of iterations in this mode.

Our proposed algorithm ESPO will increase the sample size when $t \in \mathcal{B}_{\text{soft}}^{\text{conf}}$ and decrease the sample size when $t \in \mathcal{B}_{\text{soft}}^{\text{no}}$. So as long as there exists at least one iteration $t^\star \in \mathcal{B}_{\text{soft}}^{\text{conf}}$ with $\left( 1 + \frac{\cos \theta_{rc}^{t^\star} \|\mathbf{g}_c\|}{\|\mathbf{g}_r\|} \right) > 1$, we can increase the $s_{t^\star}^{\text{B}}$ by $s_{\text{extra}} < s_{\text{batch}} \left( 1 + \frac{\cos \theta_{rc}^{t^\star} \|\mathbf{g}_c\|}{\|\mathbf{g}_r\|} \right)$ and decrease any $s_t^{\text{B}}$ by $s_{\text{extra}} \cdot \left( 1 + \frac{\cos \theta_{rc}^{t^\star} \|\mathbf{g}_c\|}{\|\mathbf{g}_r\|} \right)$ at time $t \in \mathcal{B}_{\text{soft}}^{\text{no}}$. Consequently, the total number of samples are smaller and (56) still holds. So we complete the proof.

## A.6 Proof of auxiliary results

### A.6.1 Proof of Lemma A.3

To begin with, note that the first two statements (15) and (16) has already been established in [53, Lemma 6]. So the remainder of the proof will focus on (17), which we recall here

$$\begin{cases} x_t^r \left( V_r^{\pi_{w_{t+1}}}(\rho) - V_r^{\pi_{w_t}}(\rho) \right) + x_t^c \left( V_c^{\pi_{w_{t+1}}}(\rho) - V_c^{\pi_{w_t}}(\rho) \right) \geq x_t^r \text{diff}_t^r + x_t^c \text{diff}_t^c & \text{if} \quad t \in \mathcal{B}_{\text{soft}}^{\text{no}} \\ y_t^r \left( V_r^{\pi_{w_{t+1}}}(\rho) - V_r^{\pi_{w_t}}(\rho) \right) + y_t^c \left( V_c^{\pi_{w_{t+1}}}(\rho) - V_c^{\pi_{w_t}}(\rho) \right) \geq y_t^r \text{diff}_t^r + y_t^c \text{diff}_t^c & \text{if} \quad t \in \mathcal{B}_{\text{soft}}^{\text{conf}}. \end{cases} \tag{58}$$

Towards this, the left hand side of the first line can be written out as

$$x_t^r \left( V_r^{\pi_{w_{t+1}}}(\rho) - V_r^{\pi_{w_t}}(\rho) \right) + x_t^c \left( V_c^{\pi_{w_{t+1}}}(\rho) - V_c^{\pi_{w_t}}(\rho) \right)$$

$$= x_t^r \frac{1}{1-\gamma} \mathbb{E}_{s \sim d_\rho} \sum_{a \in \mathcal{A}} \pi_{w_{t+1}}(a|s) A_r^{\pi_{w_t}}(s, a) + x_t^c \frac{1}{1-\gamma} \mathbb{E}_{s \sim d_\rho} \sum_{a \in \mathcal{A}} \pi_{w_{t+1}}(a|s) A_c^{\pi_{w_t}}(s, a)$$

$$= \frac{1}{1-\gamma} \mathbb{E}_{s \sim d_\rho} \sum_{a \in \mathcal{A}} \pi_{w_{t+1}}(a|s) \left( x_t^r Q_r^{\pi_{w_t}}(s, a) + x_t^r Q_c^{\pi_{w_t}}(s, a) \right)$$

$$\quad - \frac{1}{1-\gamma} \mathbb{E}_{s \sim d_\rho} \left( x_t^r V_r^{\pi_{w_t}}(s) + x_t^c V_c^{\pi_{w_t}}(s) \right)$$

$$= \frac{1}{1-\gamma} \mathbb{E}_{s \sim d_\rho} \sum_{a \in \mathcal{A}} \pi_{w_{t+1}}(a|s) \left( x_t^r \overline{Q}_t^r(s, a) + x_t^c \overline{Q}_t^c(s, a) \right)$$

$$\quad - \frac{1}{1-\gamma} \mathbb{E}_{s \sim d_\rho} \left( x_t^r V_r^{\pi_{w_t}}(s) + x_t^c V_c^{\pi_{w_t}}(s) \right)$$

$$+ \frac{1}{1-\gamma} \mathbb{E}_{s \sim d_\rho} \sum_{a \in \mathcal{A}} \pi_{w_{t+1}}(a|s) \left[ x_t^r \left( Q_r^{\pi_{w_t}}(s,a) - \bar{Q}_t^i(s,a) \right) + x_t^c \left( Q_c^{\pi_{w_t}}(s,a) - \bar{Q}_t^c(s,a) \right) \right]$$

$$\overset{(i)}{=} \frac{1}{\eta} \mathbb{E}_{s \sim d_\rho} \sum_{a \in \mathcal{A}} \pi_{w_{t+1}}(a|s) \log \left( \frac{\pi_{w_{t+1}}(a|s) Z_t^{r,c,1}(s)}{\pi_{w_t}(a|s)} \right) - \frac{1}{1-\gamma} \mathbb{E}_{s \sim d_\rho} \left( x_t^r V_r^{\pi_{w_t}}(s) + x_t^c V_c^{\pi_{w_t}}(s) \right)$$

$$+ \frac{1}{1-\gamma} \mathbb{E}_{s \sim d_\rho} \sum_{a \in \mathcal{A}} \pi_{w_{t+1}}(a|s) \left[ x_t^r \left( Q_r^{\pi_{w_t}}(s,a) - \bar{Q}_t^i(s,a) \right) + x_t^c \left( Q_c^{\pi_{w_t}}(s,a) - \bar{Q}_t^c(s,a) \right) \right]$$

$$= \frac{1}{\eta} \mathbb{E}_{s \sim d_\rho} D_{\mathrm{KL}}(\pi_{w_{t+1}} || \pi_{w_t}) + \frac{1}{\eta} \mathbb{E}_{s \sim d_\rho} \log Z_t^{r,c,1}(s) - \frac{1}{1-\gamma} \mathbb{E}_{s \sim d_\rho} \left( x_t^r V_r^{\pi_{w_t}}(s) + x_t^c V_c^{\pi_{w_t}}(s) \right)$$

$$+ \frac{1}{1-\gamma} \mathbb{E}_{s \sim d_\rho} \sum_{a \in \mathcal{A}} \pi_{w_{t+1}}(a|s) \left[ x_t^r \left( Q_r^{\pi_{w_t}}(s,a) - \bar{Q}_t^i(s,a) \right) + x_t^c \left( Q_c^{\pi_{w_t}}(s,a) - \bar{Q}_t^c(s,a) \right) \right],$$

$$(59)$$

where $(i)$ follows from the update rule in Lemma A.2. To continue, invoking the basic fact $D_{\mathrm{KL}}(\cdot \,|\, \cdot) \geq 0$, we have

$$x_t^r \left( V_r^{\pi_{w_{t+1}}}(\rho) - V_r^{\pi_{w_t}}(\rho) \right) + x_t^c \left( V_c^{\pi_{w_{t+1}}}(\rho) - V_c^{\pi_{w_t}}(\rho) \right)$$

$$\geq \frac{1}{\eta} \mathbb{E}_{s \sim d_\rho} \left( \log Z_t^{r,c,1}(s) - \frac{\eta}{1-\gamma} \mathbb{E}_{s \sim d_\rho} \left( x_t^r V_r^{\pi_{w_t}}(s) + x_t^c V_c^{\pi_{w_t}}(s) \right) \right.$$

$$\left. + \frac{\eta}{1-\gamma} \sum_{a \in \mathcal{A}} \pi_{w_t}(a|s) \left[ x_t^r \left| Q_r^{\pi_{w_t}}(s,a) - \bar{Q}_t^i(s,a) \right| + x_t^c \left| Q_c^{\pi_{w_t}}(s,a) - \bar{Q}_t^c(s,a) \right| \right] \right)$$

$$- \frac{1}{1-\gamma} \mathbb{E}_{s \sim d_\rho} \sum_{a \in \mathcal{A}} \pi_{w_t}(a|s) \left[ x_t^r \left| Q_r^{\pi_{w_t}}(s,a) - \bar{Q}_t^i(s,a) \right| + x_t^c \left| Q_c^{\pi_{w_t}}(s,a) - \bar{Q}_t^c(s,a) \right| \right]$$

$$- \frac{1}{1-\gamma} \mathbb{E}_{s \sim d_\rho} \sum_{a \in \mathcal{A}} \pi_{w_{t+1}}(a|s) \left[ x_t^r \left| Q_r^{\pi_{w_t}}(s,a) - \bar{Q}_t^i(s,a) \right| + x_t^c \left| Q_c^{\pi_{w_t}}(s,a) - \bar{Q}_t^c(s,a) \right| \right]$$

$$\geq \frac{1-\gamma}{\eta} \mathbb{E}_{s \sim \rho} \left( \log Z_t^{r,c,1}(s) - \frac{\eta}{1-\gamma} \mathbb{E}_{s \sim d_\rho} \left( x_t^r V_r^{\pi_{w_t}}(s) + x_t^c V_c^{\pi_{w_t}}(s) \right) \right.$$

$$\left. + \frac{\eta}{1-\gamma} \sum_{a \in \mathcal{A}} \pi_{w_t}(a|s) \left[ x_t^r \left| Q_r^{\pi_{w_t}}(s,a) - \bar{Q}_t^i(s,a) \right| + x_t^c \left| Q_c^{\pi_{w_t}}(s,a) - \bar{Q}_t^c(s,a) \right| \right] \right)$$

$$- \frac{1}{1-\gamma} \mathbb{E}_{s \sim d_\rho} \sum_{a \in \mathcal{A}} \pi_{w_t}(a|s) \left[ x_t^r \left| Q_r^{\pi_{w_t}}(s,a) - \bar{Q}_t^i(s,a) \right| + x_t^c \left| Q_c^{\pi_{w_t}}(s,a) - \bar{Q}_t^c(s,a) \right| \right]$$

$$- \frac{1}{1-\gamma} \mathbb{E}_{s \sim d_\rho} \sum_{a \in \mathcal{A}} \pi_{w_{t+1}}(a|s) \left[ x_t^r \left| Q_r^{\pi_{w_t}}(s,a) - \bar{Q}_t^i(s,a) \right| + x_t^c \left| Q_c^{\pi_{w_t}}(s,a) - \bar{Q}_t^c(s,a) \right| \right]$$

$$= x_t^r \mathsf{diff}_t^r + x_t^c \mathsf{diff}_t^c \qquad (60)$$

where the penultimate inequality holds by the fact $\|d_\rho/\rho\|_\infty \geq 1 - \gamma$ and the following claim which will be proved momentarily:

$$\log Z_t^{r,c,1}(s) - \frac{\eta}{1-\gamma} \mathbb{E}_{s \sim d_\rho} \left( x_t^r V_r^{\pi_{w_t}}(s) + x_t^c V_c^{\pi_{w_t}}(s) \right)$$

$$+ \frac{\eta}{1-\gamma} \sum_{a \in \mathcal{A}} \pi_{w_t}(a|s) \left[ x_t^r \left| Q_r^{\pi_{w_t}}(s,a) - \bar{Q}_t^i(s,a) \right| + x_t^c \left| Q_c^{\pi_{w_t}}(s,a) - \bar{Q}_t^c(s,a) \right| \right] \geq 0. \quad (61)$$

So the rest of the proof is to verify (61). To do so, applying the definition of $Z_t^{r,c,1}$ in (13), we observe that

$$\log Z_t^{r,c,1}(s) - \frac{\eta}{1-\gamma} \mathbb{E}_{s \sim d_\rho} \left( x_t^r V_r^{\pi_{w_t}}(s) + x_t^c V_c^{\pi_{w_t}}(s) \right)$$

$$= \log \left( \sum_{a \in \mathcal{A}} \pi_{w_t}(a|s) \exp \left( \frac{\eta \left( x_t^r \bar{Q}_t^r(s,a) + x_t^c \bar{Q}_t^c(s,a) \right)}{(1-\gamma)} \right) \right)$$

$$-\frac{\eta}{1-\gamma}\mathbb{E}_{s\sim d_\rho}\left(x_t^r V_r^{\pi_{w_t}}(s) + x_t^c V_c^{\pi_{w_t}}(s)\right)$$

$$\geq \sum_{a\in\mathcal{A}} \pi_{w_t}(a|s)\frac{\eta\left(x_t^r\bar{Q}_t^r(s,a) + x_t^c\bar{Q}_t^c(s,a)\right)}{(1-\gamma)} - \frac{\eta}{1-\gamma}\mathbb{E}_{s\sim d_\rho}\left(x_t^r V_r^{\pi_{w_t}}(s) + x_t^c V_c^{\pi_{w_t}}(s)\right)$$

$$= \sum_{a\in\mathcal{A}} \pi_{w_t}(a|s)\frac{\eta}{1-\gamma}\left[x_t^r\left(\bar{Q}_t^r(s,a) - Q_r^{\pi_{w_t}}(s,a)\right) + x_t^c\left(\bar{Q}_t^c(s,a) - Q_c^{\pi_{w_t}}(s,a)\right)\right]$$

$$+ \sum_{a\in\mathcal{A}} \pi_{w_t}(a|s)\frac{\eta}{1-\gamma}\left(x_t^r Q_r^{\pi_{w_t}}(s,a) + x_t^c Q_c^{\pi_{w_t}}(s,a)\right) - \frac{\eta}{1-\gamma}\mathbb{E}_{s\sim d_\rho}\left(x_t^r V_r^{\pi_{w_t}}(s) + x_t^c V_c^{\pi_{w_t}}(s)\right)$$

$$= \sum_{a\in\mathcal{A}} \pi_{w_t}(a|s)\frac{\eta}{1-\gamma}\left[x_t^r\left(\bar{Q}_t^r(s,a) - Q_r^{\pi_{w_t}}(s,a)\right) + x_t^c\left(\bar{Q}_t^c(s,a) - Q_c^{\pi_{w_t}}(s,a)\right)\right], \qquad (62)$$

which complete the proof of the first line of (17). The second line of (17) can be proved analogously.

### A.6.2 Proof of Lemma A.4

First, the first two statements (65) and (68) have already been established in [53, Lemma 7]. So we focus on (17) throughout this subsection.

Consider the first line of (17), applying Lemma (A.1) and following the pipeline for (59) yields

$$x_t^r\left(V_r^{\pi^*}(\rho) - V_r^{\pi_{w_t}}(\rho)\right) + x_t^c\left(V_c^{\pi^*}(\rho) - V_c^{\pi_{w_t}}(\rho)\right)$$

$$= \frac{1}{\eta}\mathbb{E}_{s\sim d^\star}\left(D_{\text{KL}}(\pi^*||\pi_{w_t}) - D_{\text{KL}}(\pi^*||\pi_{w_{t+1}})\right) + \frac{1}{\eta}\mathbb{E}_{s\sim d^\star}\log Z_t^{r,c,1}(s)$$

$$- \frac{1}{1-\gamma}\mathbb{E}_{s\sim d^\star}\left(x_t^r V_r^{\pi_{w_t}}(s) + x_t^c V_c^{\pi_{w_t}}(s)\right)$$

$$+ \frac{1}{1-\gamma}\mathbb{E}_{s\sim d^\star}\sum_{a\in\mathcal{A}}\pi^\star(a|s)\left[x_t^r\left(Q_r^{\pi_{w_t}}(s,a) - \bar{Q}_t^i(s,a)\right) + x_t^c\left(Q_c^{\pi_{w_t}}(s,a) - \bar{Q}_t^c(s,a)\right)\right]$$

$$\leq \frac{1}{\eta}\mathbb{E}_{s\sim d^\star}\left(D_{\text{KL}}(\pi^*||\pi_{w_t}) - D_{\text{KL}}(\pi^*||\pi_{w_{t+1}})\right)$$

$$+ \frac{1}{\eta}\mathbb{E}_{s\sim d^\star}\left(\log Z_t^{r,c,1}(s) - \frac{\eta}{1-\gamma}\mathbb{E}_{s\sim d_\rho}\left(x_t^r V_r^{\pi_{w_t}}(s) + x_t^c V_c^{\pi_{w_t}}(s)\right)\right.$$

$$\left.+ \frac{\eta}{1-\gamma}\sum_{a\in\mathcal{A}}\pi_{w_t}(a|s)\left[x_t^r\left|Q_r^{\pi_{w_t}}(s,a) - \bar{Q}_t^i(s,a)\right| + x_t^c\left|Q_c^{\pi_{w_t}}(s,a) - \bar{Q}_t^c(s,a)\right|\right]\right)$$

$$+ \frac{1}{1-\gamma}\mathbb{E}_{s\sim d^\star}\sum_{a\in\mathcal{A}}\pi^\star(a|s)\left[x_t^r\left(Q_r^{\pi_{w_t}}(s,a) - \bar{Q}_t^i(s,a)\right) + x_t^c\left(Q_c^{\pi_{w_t}}(s,a) - \bar{Q}_t^c(s,a)\right)\right]$$

$$\overset{(i)}{\leq} \frac{1}{\eta}\mathbb{E}_{s\sim d^\star}\left(D_{\text{KL}}(\pi^*||\pi_{w_t}) - D_{\text{KL}}(\pi^*||\pi_{w_{t+1}})\right)$$

$$+ \frac{1}{1-\gamma}\left[x_t^r\left(V_r^{\pi_{w_{t+1}}}(\rho) - V_r^{\pi_{w_t}}(\rho)\right) + x_t^c\left(V_c^{\pi_{w_{t+1}}}(\rho) - V_c^{\pi_{w_t}}(\rho)\right)\right]$$

$$+ \frac{1}{(1-\gamma)^2}\mathbb{E}_{s\sim d_\rho}\sum_{a\in\mathcal{A}}\pi_{w_t}(a|s)\left[x_t^r\left|Q_r^{\pi_{w_t}}(s,a) - \bar{Q}_t^i(s,a)\right| + x_t^c\left|Q_c^{\pi_{w_t}}(s,a) - \bar{Q}_t^c(s,a)\right|\right]$$

$$+ \frac{1}{(1-\gamma)^2}\mathbb{E}_{s\sim d_\rho}\sum_{a\in\mathcal{A}}\pi_{w_{t+1}}(a|s)\left[x_t^r\left|Q_r^{\pi_{w_t}}(s,a) - \bar{Q}_t^i(s,a)\right| + x_t^c\left|Q_c^{\pi_{w_t}}(s,a) - \bar{Q}_t^c(s,a)\right|\right]$$

$$+ \frac{1}{1-\gamma}\mathbb{E}_{s\sim d^\star}\sum_{a\in\mathcal{A}}\pi^\star(a|s)\left[x_t^r\left(Q_r^{\pi_{w_t}}(s,a) - \bar{Q}_t^i(s,a)\right) + x_t^c\left(Q_c^{\pi_{w_t}}(s,a) - \bar{Q}_t^c(s,a)\right)\right]$$

$$\leq \frac{1}{\eta}\mathbb{E}_{s\sim d^\star}\left(D_{\text{KL}}(\pi^*||\pi_{w_t}) - D_{\text{KL}}(\pi^*||\pi_{w_{t+1}})\right) + \frac{2v_{\max}}{(1-\gamma)^2}\|w_{t+1} - w_t\|_2$$

$$+ \frac{3}{(1-\gamma)^2}\left[x_t^r\left\|Q_r^{\pi_{w_t}}(s,a) - \bar{Q}_t^i(s,a)\right\|_2 + x_t^c\left\|Q_c^{\pi_{w_t}}(s,a) - \bar{Q}_t^c(s,a)\right\|_2\right]$$

$$\leq \frac{1}{\eta}\mathbb{E}_{s\sim d^\star}(D_{\text{KL}}(\pi^*||\pi_{w_t}) - D_{\text{KL}}(\pi^*||\pi_{w_{t+1}})) + \frac{2\eta v_{\max}^2|\mathcal{S}||\mathcal{A}|}{(1-\gamma)^3}$$

$$+ \frac{3(1+\eta v_{\max})}{(1-\gamma)^2}\left[x_t^r\left\|Q_r^{\pi_{w_t}}(s,a) - \bar{Q}_t^i(s,a)\right\|_2 + x_t^c\left\|Q_c^{\pi_{w_t}}(s,a) - \bar{Q}_t^c(s,a)\right\|_2\right], \tag{63}$$

where (i) holds by applying Lemma A.3, the penultimate inequality holds by the Lipschitz property of $V_r^{\pi_w}(\rho)$ and $V_c^{\pi_w}(\rho)$, and the last inequality can be verified following the last line in the proof of [53, Lemma 7].

Similarly, we have

$$y_t^r\left(V_r^{\pi^*}(\rho) - V_r^{\pi_{w_t}}(\rho)\right) + y_t^c\left(V_c^{\pi^*}(\rho) - V_c^{\pi_{w_t}}(\rho)\right)$$

$$\leq \frac{1}{\eta}\mathbb{E}_{s\sim d^\star}(D_{\text{KL}}(\pi^*||\pi_{w_t}) - D_{\text{KL}}(\pi^*||\pi_{w_{t+1}})) + \frac{2\eta v_{\max}^2(y_t^r + y_t^c)|\mathcal{S}||\mathcal{A}|}{(1-\gamma)^3}$$

$$+ \frac{3(1+\eta v_{\max})}{(1-\gamma)^2}\left[x_t^r\left\|Q_r^{\pi_{w_t}}(s,a) - \bar{Q}_t^i(s,a)\right\|_2 + x_t^c\left\|Q_c^{\pi_{w_t}}(s,a) - \bar{Q}_t^c(s,a)\right\|_2\right], \tag{64}$$

which complete the proof.

### A.6.3 Proof of Lemma A.5

Invoking Lemma (A.4) for the four modes when $t \in \mathcal{B}_r$, $t \in \mathcal{B}_{\text{soft}}^{\text{no}}$, $t \in \mathcal{B}_{\text{soft}}^{\text{conf}}$, and $t \in \mathcal{B}_c$ and summing up them together for $t = 1, 2, \cdots, T$ yields

$$\eta \sum_{t\in\mathcal{B}_r}(V_r^{\pi^*}(\rho) - V_r^{\pi_{w_t}}(\rho)) + \eta \sum_{t\in\mathcal{B}_{\text{soft}}^{\text{no}}}\left[x_r^t(V_r^{\pi_{w_t}}(\rho) - V_r^{\pi^*}(\rho)) + x_c^t(V_c^{\pi^*}(\rho) - V_c^{\pi_{w_t}}(\rho))\right]$$

$$+ \eta \sum_{t\in\mathcal{B}_{\text{soft}}^{\text{conf}}}\left[y_r^t(V_r^{\pi_{w_t}}(\rho) - V_r^{\pi^*}(\rho)) + y_c^t(V_c^{\pi^*}(\rho) - V_c^{\pi_{w_t}}(\rho))\right] + \eta \sum_{t\in\mathcal{B}_c}(V_r^{\pi_{w_t}}(\rho) - V_r^{\pi^*}(\rho))$$

$$\leq \mathbb{E}_{s\sim d^\star}D_{\text{KL}}(\pi^*||\pi_{w_0}) + \frac{2\eta^2 v_{\max}^2|\mathcal{S}||\mathcal{A}|}{(1-\gamma)^3}\left[(T - |\mathcal{B}_{\text{soft}}^{\text{conf}}|) + \sum_{t\in\mathcal{B}_{\text{soft}}^{\text{conf}}}(y_t^c + y_t^r)\right] + \frac{3\eta(1+\eta v_{\max})}{(1-\gamma)^2}\epsilon_{\text{pi}}, \tag{65}$$

where $\epsilon_{\text{pi}}$ is defined in (24).

Then we consider several different modes separately:

- When $t \in \mathcal{B}_c$: we have $\overline{V}_t^c > b + h^+$, which indicates that

$$V_c^{\pi_{w_t}}(\rho) - V_c^{\pi^*}(\rho)$$
$$= \overline{V}_t^c(\rho) - V_c^{\pi^*}(\rho) + V_c^{\pi_{w_t}}(\rho) - \overline{V}_t^c \geq h^+ - |V_c^{\pi_{w_t}}(\rho) - \overline{V}_t^c| \geq h^+ - \|Q_c(\pi_{w_t}) - \overline{Q}_t^c\|_2. \tag{66}$$

- when $t \in \mathcal{B}_{\text{soft}}$: $\overline{V}_t^c \geq b - h^-$, one has

$$V_c^{\pi_{w_t}}(\rho) - V_c^{\pi^*}(\rho)$$
$$= \overline{V}_t^{(}\rho) - V_c^{\pi^*}(\rho) + V_c^{\pi_{w_t}}(\rho) - \overline{V}_t^c \geq -h^- - |V_c^{\pi_{w_t}}(\rho) - \overline{V}_t^c| \geq -h^- - \|Q_c(\pi_{w_t}) - \overline{Q}_t^c\|_2. \tag{67}$$

Summing up the above two modes and plugging them back to (65) leads to

$$\eta \sum_{t\in\mathcal{B}_r}(V_r^{\pi^*}(\rho) - V_r^{\pi_{w_t}}(\rho)) + \eta \sum_{t\in\mathcal{B}_{\text{soft}}^{\text{no}}}x_r^t(V_r^{\pi_{w_t}}(\rho) - V_r^{\pi^*}(\rho)) + \eta \sum_{t\in\mathcal{B}_{\text{soft}}^{\text{conf}}}y_r^t(V_r^{\pi_{w_t}}(\rho) - V_r^{\pi^*}(\rho))$$

$$+ \eta h^+|\mathcal{B}_c| - \eta h^-\sum_{t\in\mathcal{B}_{\text{soft}}^{\text{no}}}x_r^t - \eta h^-\sum_{t\in\mathcal{B}_{\text{soft}}^{\text{conf}}}y_r^t$$

$$\leq \mathbb{E}_{s \sim d^*} D_{\mathsf{KL}}(\pi^* || \pi_{w_0}) + \frac{2\eta^2 v_{\max}^2 |\mathcal{S}||\mathcal{A}|}{(1-\gamma)^3} \left[ (T - |\mathcal{B}_{\mathsf{soft}}^{\mathsf{conf}}|) + \sum_{t \in \mathcal{B}_{\mathsf{soft}}^{\mathsf{conf}}} (y_t^c + y_t^r) \right] + \frac{3\eta(1 + \eta v_{\max})}{(1-\gamma)^2} \epsilon_{\mathsf{pi}}$$

$$(68)$$

To continue, invoking [53, Lemma 2] leads to when the iterations of policy evaluation obey $T_{\mathsf{pi}} = \widetilde{O}\big(\frac{T \log(\frac{|\mathcal{S}||\mathcal{A}|}{\delta})}{(1-\gamma)^3 |\mathcal{S}||\mathcal{A}|}\big)$. With probability at least $1 - \delta$, we have for all $1 \leq t \leq T$,

$$\big\| Q_r^{\pi_{w_t}} - \bar{Q}_t^r \big\|_2 \leq \frac{1}{2} \sqrt{\frac{(1-\gamma)|\mathcal{S}||\mathcal{A}|}{T}}$$

$$\text{and} \quad \big\| Q_c^{\pi_{w_t}} - \bar{Q}_t^c \big\|_2 \leq \frac{1}{2} \sqrt{\frac{(1-\gamma)|\mathcal{S}||\mathcal{A}|}{T}} \leq \sqrt{\frac{(1-\gamma)|\mathcal{S}||\mathcal{A}|}{T}}. \quad (69)$$

Combining this fact with the definition in (24) directly leads to

$$\epsilon_{\mathsf{pi}} = \sum_{t \in \mathcal{B}_r} \big\| Q_r^{\pi_{w_t}} - \bar{Q}_t^r \big\|_2 + \sum_{t \in \mathcal{B}_c} \big\| Q_c^{\pi_{w_t}} - \bar{Q}_t^c \big\|_2 + \sum_{t \in \mathcal{B}_{\mathsf{soft}}^{\mathsf{no}}} \big( x_t^r \| Q_r^{\pi_{w_t}} - \bar{Q}_t^r \|_2 + x_t^c \| Q_c^{\pi_{w_t}} - \bar{Q}_t^c \|_2 \big)$$

$$+ \sum_{t \in \mathcal{B}_{\mathsf{soft}}^{\mathsf{conf}}} \big( y_t^r \| Q_r^{\pi_{w_t}} - \bar{Q}_t^r \|_2 + y_t^c \| Q_c^{\pi_{w_t}} - \bar{Q}_t^c \|_2 \big)$$

$$\leq \sqrt{(1-\gamma)|\mathcal{S}||\mathcal{A}|T}. \quad (70)$$

Plugging (70) back into (68) complete the proof:

$$\eta \sum_{t \in \mathcal{B}_r} (V_r^{\pi^*}(\rho) - V_r^{\pi_{w_t}}(\rho)) + \eta \sum_{t \in \mathcal{B}_{\mathsf{soft}}^{\mathsf{no}}} x_r^t (V_r^{\pi_{w_t}}(\rho) - V_r^{\pi^*}(\rho)) + \eta \sum_{t \in \mathcal{B}_{\mathsf{soft}}^{\mathsf{conf}}} y_r^t (V_r^{\pi_{w_t}}(\rho) - V_r^{\pi^*}(\rho))$$

$$+ \eta h^+ |\mathcal{B}_c| - \eta h^- \sum_{t \in \mathcal{B}_{\mathsf{soft}}^{\mathsf{no}}} x_r^t - \eta h^- \sum_{t \in \mathcal{B}_{\mathsf{soft}}^{\mathsf{conf}}} y_r^t$$

$$\leq \mathbb{E}_{s \sim d^*} D_{\mathsf{KL}}(\pi^* || \pi_{w_0}) + \frac{2\eta^2 v_{\max}^2 |\mathcal{S}||\mathcal{A}|}{(1-\gamma)^3} \left[ (T - |\mathcal{B}_{\mathsf{soft}}^{\mathsf{conf}}|) + \sum_{t \in \mathcal{B}_{\mathsf{soft}}^{\mathsf{conf}}} (y_t^c + y_t^r) \right] + \frac{3\eta(1 + \eta v_{\max})}{(1-\gamma)^2} \epsilon_{\mathsf{pi}}$$

$$\leq \mathbb{E}_{s \sim d^*} D_{\mathsf{KL}}(\pi^* || \pi_{w_0}) + \frac{4\eta^2 v_{\max}^2 |\mathcal{S}||\mathcal{A}|T}{(1-\gamma)^3} + \frac{3\eta(1 + \eta v_{\max})\sqrt{|\mathcal{S}||\mathcal{A}|T}}{(1-\gamma)^{1.5}} \quad (71)$$

since $(y_t^c + y_t^r) \leq 2$.

### A.6.4 Proof of Lemma A.6

The first claim is easily verified since if $\mathcal{B}_r \cup \mathcal{B}_{\mathsf{soft}} = \emptyset$, then $|\mathcal{B}_c| = T$. Applying Lemma A.5 gives

$$\eta h^+ |\mathcal{B}_c| = \eta h^+ T \leq \mathbb{E}_{s \sim d^*} D_{\mathsf{KL}}(\pi^* || \pi_{w_0}) + \frac{4\eta^2 v_{\max}^2 |\mathcal{S}||\mathcal{A}|T}{(1-\gamma)^3} + \frac{3\eta(1 + \eta v_{\max})\sqrt{|\mathcal{S}||\mathcal{A}|T}}{(1-\gamma)^{1.5}},$$

$$(72)$$

which contradict with the assumption (25). So we have $\mathcal{B}_r \cup \mathcal{B}_{\mathsf{soft}} \neq \emptyset$.

Then the rest of the proof focus on the second claim. Towards this, if

$$\sum_{t \in \mathcal{B}_r} (V_r^{\pi^*}(\rho) - V_r^{\pi_{w_t}}(\rho)) + \sum_{t \in \mathcal{B}_{\mathsf{soft}}^{\mathsf{no}}} x_r^t (V_r^{\pi^*}(\rho) - V_r^{\pi_{w_t}}(\rho)) + \sum_{t \in \mathcal{B}_{\mathsf{soft}}^{\mathsf{conf}}} y_r^t (V_r^{\pi^*}(\rho) - V_r^{\pi_{w_t}}(\rho)) \leq 0,$$

$$(73)$$

then the condition (b) holds. Otherwise, applying Lemma A.5 yields

$$\eta h^+ |\mathcal{B}_c| - \eta h^- \sum_{t \in \mathcal{B}_{\mathsf{soft}}^{\mathsf{no}}} x_r^t - \eta h^- \sum_{t \in \mathcal{B}_{\mathsf{soft}}^{\mathsf{conf}}} y_r^t$$

$$\leq \mathbb{E}_{s \sim d^*} D_{\mathsf{KL}}(\pi^* || \pi_{w_0}) + \frac{4\eta^2 v_{\max}^2 |\mathcal{S}||\mathcal{A}|T}{(1-\gamma)^3} + \frac{3\eta(1 + \eta v_{\max})\sqrt{|\mathcal{S}||\mathcal{A}|T}}{(1-\gamma)^{1.5}} \quad (74)$$

Then if $|\mathcal{B}_r \cup \mathcal{B}_{\mathsf{soft}}| < T/2$, we have $|\mathcal{B}_{\mathsf{c}}| \geq \frac{T}{2}$ and thus

$$\frac{\eta h^+ T}{2} - \eta h^- T \leq \eta h^+ |\mathcal{B}_{\mathsf{c}}| - 2(T - |\mathcal{B}_{\mathsf{c}}|)\eta h^- \leq \eta h^+ |\mathcal{B}_{\mathsf{c}}| - \eta h^- \sum_{t \in \mathcal{B}_{\mathsf{soft}}^{\mathsf{no}}} 1 - \eta h^- \sum_{t \in \mathcal{B}_{\mathsf{soft}}^{\mathsf{conf}}} 2$$

$$\leq \eta h^+ |\mathcal{B}_{\mathsf{c}}| - \eta h^- \sum_{t \in \mathcal{B}_{\mathsf{soft}}^{\mathsf{no}}} x_r^t - \eta h^- \sum_{t \in \mathcal{B}_{\mathsf{soft}}^{\mathsf{conf}}} y_r^t$$

$$\leq \mathbb{E}_{s \sim d^*} D_{\mathrm{KL}}(\pi^* || \pi_{w_0}) + \frac{4\eta^2 v_{\max}^2 |\mathcal{S}||\mathcal{A}|T}{(1-\gamma)^3} + \frac{3\eta(1 + \eta v_{\max})\sqrt{|\mathcal{S}||\mathcal{A}|T}}{(1-\gamma)^{1.5}}, \tag{75}$$

which yields

$$\frac{\eta h^+ T}{2} \leq \mathbb{E}_{s \sim d^*} D_{\mathrm{KL}}(\pi^* || \pi_{w_0}) + \frac{4\eta^2 v_{\max}^2 |\mathcal{S}||\mathcal{A}|T}{(1-\gamma)^3} + \frac{3\eta(1 + \eta v_{\max})\sqrt{|\mathcal{S}||\mathcal{A}|T}}{(1-\gamma)^{1.5}} \tag{76}$$

that is contradict with the assumption (25).

# B  Practical Algorithm

---

**Algorithm 1 ESPO**: Improving Efficiency of Safe Policy Optimization.

---

1: **Inputs**: initial policy with parameters $\pi_{w_0}$, positive slack value $h_t^+ \in [0, +\infty)$, negative slack value $h_t^- \in (-\infty, 0]$, the cost value as $V_{c_t}^{\pi_{w_0}}(\rho)$ at step $t$, the cost limit as $b$, positive sample penalty $\zeta^+ \in [0, +\infty)$, negative sample penalty $\zeta^- \in (-1, 0]$, gradient angles $\theta_{r,c}$, sample size $X$.

2: **for** $t = 0, \ldots, T-1$ **do**

3:     **if** $h^+$ iteratively decreases **then**

4:         $h_t^+ \leftarrow h_t^+ - h_t^+/T$

5:     **end if**

6:     **if** $h_t^-$ iteratively increases **then**

7:         $h_t^- \leftarrow h_t^- - h_t^-/T$

8:     **end if**

9:     **if** $\zeta_t^+$ iteratively decreases **then**

10:        $\zeta_t^+ \leftarrow \zeta_t^+ - \zeta_t^+/T$

11:     **end if**

12:     **if** $\zeta_t^-$ iteratively increases **then**

13:        $\zeta_t^- \leftarrow \zeta_t^- - \zeta_t^-/T$

14:     **end if**

15:     **if** $V_{c_t}^{\pi_{w_t}}(\rho) > (h_t^+ + b)$ **then**

16:        Adjust sample size $X_t$ with Equation (7).

17:        Update policy $\pi_{w_t}$ to ensure safety with Equation (2).

18:     **else if** $(h_t^- + b) \leq V_{c_t}^{\pi_{w_t}}(\rho) \leq (h_t^+ + b)$ **then**

19:        **if** For gradients $\mathbf{g}_r$ and $\mathbf{g}_c$, $\theta_{r,c} \leq 90°$ **then**

20:           Adjust sample size $X_t$ with Equation (7).

21:           Update the policy $\pi_{w_t}$ with Equation (3).

22:        **else**

23:           Adjust sample size $X_t$ with Equation (6).

24:           Update the policy $\pi_{w_t}$ with Equation (4).

25:        **end if**

26:     **else if** $V_{c_t}^{\pi_{w_t}}(\rho) < (h_t^- + b)$ **then**

27:        Adjust sample size $X_t$ with Equation (7).

28:        Update policy $\pi_{w_t}$ to maximize reward $V_{r,t}^{\pi_{w_t}}(\rho)$ with Equation (5).

29:     **end if**

30:     Policy evaluation under $\pi_{w_t}$ involves estimating the values of rewards and constraints.

31:     Sample pairs $(s_j, a_j)$ from the buffer $\mathcal{B}_t$ according to the distribution $\rho \cdot \pi_{w_t}$ and compute the estimation $V_{r,t}^{\pi_{w_t}}(\rho)$ and $V_{c_t}^{\pi_{w_t}}(\rho)$, where $s_j$ represents the state and $a_j$ represents the action, $j$ is is the index for the sampled pairs.

32: **end for**

33: **Outputs**: $\pi_{w_t}$.

---

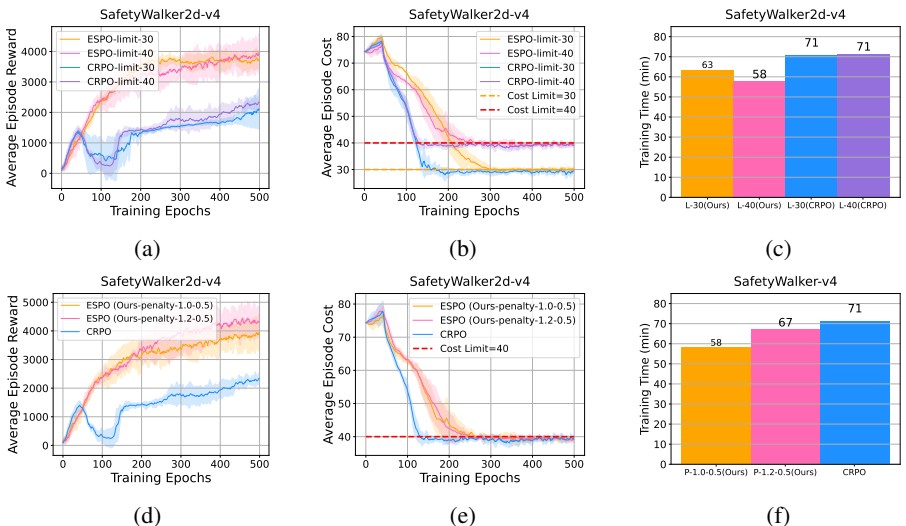

Figure 4: Ablation experiments: Experiments of different cost limits and sample sizes.

## C  Ablation Experiments

To further evaluate the effectiveness of our method, we conduct a series of ablation experiments regarding different cost limits, different sample sizes, learning rates, gradient weights, and update style analysis. These ablation experiments are instrumental in providing a deeper insight into our method, shedding light on its strengths and potential areas for improvement. Through this rigorous evaluation, we aim to substantiate the adaptability of our method, ensuring its applicability and effectiveness in a wide range of safe RL scenarios.

**Different Cost Limits:** As depicted in Figures 4(a)-(c), we evaluate our method on the *SafetyWalker2d-v4* tasks under different cost limits, maintaining identical sample manipulation settings. Our method exhibits similar reward performance at cost limits of $30$ and $40$. This similarity in performance is attributed to our method's capacity to dynamically adjust the sample size, a critical factor in optimizing for reward maximization while ensuring safety. Moreover, the training time for the task with a cost limit of $30$ is $63$ minutes, slightly longer than the $58$ minutes required for the limit of $40$. This observation can be explained by the increased challenge and larger conflict between reward and safety presented at the lower constraint limit of $30$, necessitating a more significant number of samples for effective optimization. Notably, our method can ensure safety across these various constraint-limited tasks and outperforms CRPO in reward performance and training efficiency.

**Different Sample Sizes:** As illustrated in Figures 4(d)-(f), we conduct an assessment of our method on the *SafetyWalker2d-v4* tasks, exploring different sample sizes while keeping the cost limit settings constant. In these experiments, we compare the outcomes of using sample sizes set at $1.2X$ and $0.5X$ against $1.0X$ and $0.5X$. Notably, both settings successfully ensured safety. On the one hand, the reward performance achieved with a sample size of $1.2X$ and $0.5X$ surpasses that of $1.0X$ and $0.5X$, indicating the effectiveness of larger sample size in enhancing performance; on the other hand, the training time for the sample size of $1.2X$ and $0.5X$ is recorded at $67$ minutes, which is longer than the $58$ minutes required for the sample size of $1.0X$ and $0.5X$. Despite this increased training time, it remains less than the $71$ minutes recorded for CRPO. These results underscore the potential benefits of utilizing more samples to improve performance in safe RL tasks. Importantly, in both sample manipulation settings, our method ensures safety and outperforms CRPO in terms of reward performance and training efficiency.

**Different Gradient Weights:** $x_t^r$ and $x_t^c$ represent the weight of the reward gradient (resp. the safety cost gradient) in the final gradient $w_{t+1}$. So $x_t^r + x_t^c = 1$ all the time and, for instance, $x_t^r = 1$ (resp. $x_t^c = 1$) indicates we only use reward gradient (resp. safety cost gradient), and $x_t^r = x_t^c = 0.5$ denotes reward and safety cost objectives are considered equally important in the overall gradient. In general, $x_t^r$ and $x_t^c$ are hyperparameters in the framework that we can either pre-set as a fixed value or adaptively adjust during the running process as needed. For instance, we can set $x_t^r$ to be larger if we care more about the reward performance; otherwise, we set $x_t^c$ to be

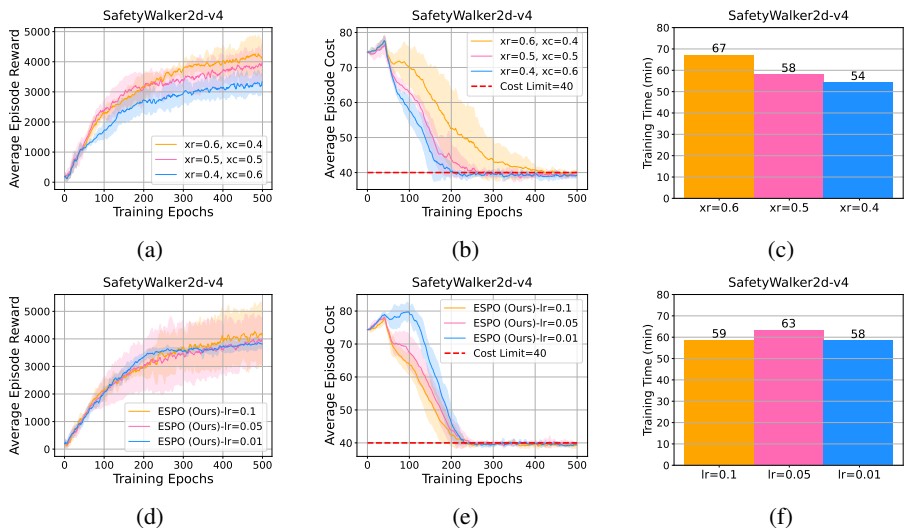

Figure 5: Ablation experiments: Experiments of different gradient weights $x^r$ and $x^c$ and different learning rates.

larger to enhance safety. Throughout our experiments, we just set $x_t^r = x_t^c = 0.5$ for simplicity, which also showed superior performance than prior arts (see Section 5). We provide an ablation study for the hyperparameters to evaluate its effect on the performance, shown in Figures 5(a)-(c). To test the sensitivity of the performance w.r.t. the hyperparameters $x_t^r$ and $x_t^c$, we carry out experiments using other two combinations of $x_t^r$ and $x_t^c$ ($x_t^r = 0.4, x_t^c = 0.6$ or $x_t^r = 0.6, x_t^c = 0.4$). The results show that our proposed ESPO performs well using such different $x_t^r, x_t^c$ settings and is even better than using $x_t^r = x_t^c = 0.5$ sometimes.

**Different Learning Rates:** We conduct ablation studies on hyperparameters — learning rates ($l_r$), shown in Figure Figures 5(d)-(f), reveal that ESPO is robust to variations in learning rates. The results demonstrate that ESPO performs well under reasonably varying learning rates.

**Update Style Analysis:** The analysis of update style in our experiments, as illustrated in Table 3 for the *SafetyHumanoidStandup-v4* task, offers insightful contrasts between our algorithm, ESPO, and CRPO method. In these experiments, we observe the following update patterns: **1)** *CRPO's Update Style*: CRPO's approach to optimization involved 178 updates focused solely on reward optimization and 322 updates dedicated to cost optimization. This distribution suggests a significant emphasis on cost optimization, indicating that CRPO struggles to manage safety constraints. **2)** *ESPO's Update Style*: ESPO, on the other hand, showed a more dynamic update pattern. It conducted 298 updates focused on reward optimization, indicating a more efficient approach toward maximizing rewards. However, unlike CRPO, ESPO engages 199 updates characterized by simultaneous optimization of both reward and cost. Additionally, 3 updates focused exclusively on optimizing cost. By optimizing both aspects simultaneously, ESPO demonstrates a novel method of navigating the complex landscape of safe RL, which may contribute to its overall efficiency and effectiveness as observed in the task performance.

| Algorithm \ Update | Reward | Reward & Cost | Cost |
|---|---|---|---|
| ESPO (ours) | 298 | 199 | 3 |
| CRPO | 178 | / | 322 |

Table 3: Update style analysis. The *Reward update* represents the number of times the algorithm updates its policy primarily focusing on maximizing rewards, the *Cost update* refers to the number of cost updates where the safety violation happens and the primary focus is on minimizing costs, the *Reward & Cost update* corresponds to the number of times the optimization of reward and cost updates are executed simultaneously.

# D  Detailed Experiments

## D.1  Additional Experiments

The results of our experimental evaluations on the *SafetyHumanoidStandup-v4* task, as depicted in Figures 6(a)-(c), show the superior performance of our algorithm, ESPO, in comparison with SOTA primal baselines, CRPO and PCRPO. Key observations from these results include: ESPO demonstrates a remarkable ability to outperform CRPO and achieve comparable performance with PCRPO in reward while ensuring safety. Another notable aspect of ESPO's performance is that our method required less time to reach convergence than these baselines. This efficiency is crucial in practical applications where time and computational resources are often limited. ESPO requires only approximately 76.5% and 74.01% of the training time that CRPO and PCRPO need, respectively, to achieve superior performance. Specifically, as depicted in Table 1, while CRPO and PCRPO utilize 8 million samples for the *SafetyHumanoidStandup-v4* task, our method requires only 5.1 million samples for the same task. This reduction in samples is a significant advantage, highlighting ESPO's effectiveness in learning efficiency.

These results from the *SafetyHumanoidStandup-v4* task further demonstrate the effectiveness of our method in safe RL environments, showcasing its potential as a reliable and efficient solution for optimizing rewards while adhering to safety constraints.

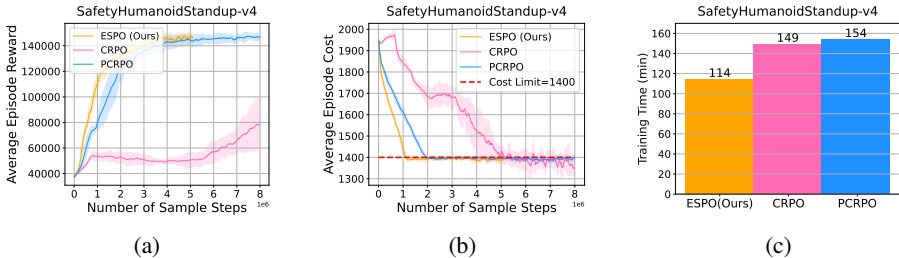

(a)                                    (b)                                    (c)

Figure 6: Performance comparisons of safe RL methods on *SafetyHumanoidStandup-v4* tasks.

## D.2  Experiment Settings

The *Safety-MuJoCo* benchmark is primarily used for primal-based methods, while the *Omnisafe* benchmark is mainly utilized for primal-dual based methods. Moreover, the *Safety-MuJoCo* benchmark is different from the *Omnisafe* benchmark in safety settings. *Safety-MuJoCo* encompasses broad safety constraints including both velocity limits and overall robot health. Accounting for multiple factors requires algorithms to consider both speed regulation and broader integrity. In contrast, the *Omnisafe* benchmark primarily focuses on robot velocity as the critical constraint. For instance, a cost of 1 is emitted whenever the robot's velocity exceeds a predefined limit. This singular focus on velocity provides a more targeted, yet still challenging, evaluation context. Through these experimental setups, we aim to comprehensively assess the effectiveness of our method in varying scenarios, ranging from the multifaceted safety constraints in *Safety-MuJoCo* to the velocity-centric constraints in *Omnisafe*. For more details, see [30] and [32]. To ensure a fair evaluation of our method's effectiveness, we conducted all experiments using at least three different random seeds.

The key parameters used in the tasks of *Safety-MuJoCo* benchmarks are provided in Table 4, Table 5 and Table 6. Note, to encourage more learning exploration, we initiate the optimization of safety after 40 epochs. Experiments in the tasks of *Safety-MuJoCo* benchmarks are conducted on a Ubuntu 20.04.3 LTS system, with an AMD Ryzen-7-2700X CPU and an NVIDIA GeForce RTX 2060 GPU.

The key parameters used on the tasks of *Omnisafe* benchmarks are provided in Table 5, Table 6, and Table 7. Experiments on the tasks of *Omnisafe* benchmarks are conducted on a Ubuntu 20.04.6 LTS system, with 2 AMD EPYC-7763 CPUs and 6 NVIDIA RTX A6000 GPUs.

| Parameters | value | Parameters | value |
|---|---|---|---|
| gamma | 0.995 | tau | 0.97 |
| l2-reg | 1e-3 | cost kl | 0.05 |
| damping | 1e-1 | batch-size | [16000, /] |
| epoch | 500 | episode length | 1000 |
| grad-c | 0.5 | neural network | MLP |
| hidden layer dim | 64 | accept ratio | 0.1 |
| energy weight | 1.0 | forward reward weight | 1.0 |

Table 4: Key parameters used in *Safety-MuJoCo* benchmarks. In ESPO, the sample size of each epoch is determined by Algorithm 1, with Equations (7) and (6), in which the $X$ is 16000.

| Tasks | $\zeta^+$ | $\zeta^-$ | Tasks | $\zeta^+$ | $\zeta^-$ |
|---|---|---|---|---|---|
| SafetyHopperVelocity-v1 | 0.1 | -0.4 | SafetyAntVelocity-v1 | 0.1 | -0.4 |
| SafetyHumanoidStandup-v4 | 0.0 | -0.5 | SafetyWalker2d-v4 | 0.0 | 0.5 |
| SafetyWalker2d-v4-a | 0.0 | -0.5 | SafetyWalker2d-v4-b | 0.2 | -0.5 |
| SafetyReacher-v4 | 0.1 | -0.3 | | | |

Table 5: Sample parameters used in *Omnisafe* and *Safety-MuJoCo* experiments. The results of *SafetyHopperVelocity-v1* and *SafetyAntVelocity-v1* are shown in Figure 3, the results of *SafetyHumanoidStandup-v4* and *SafetyWalker2d-v4* are shown in Figure 2, the results of *SafetyWalker2d-v4-a* are shown in Figures 4 (a), (b) and (c), the results of *SafetyWalker2d-v4-a* and *SafetyWalker2d-v4-b* are shown in Figures 4 (d), (e) and (f); the results of *SafetyReacher-v4* experiments are shown in Figure 2.

| Tasks | $b$ | $h^+$ | $h^-$ | Tasks | $b$ | $h^+$ | $h^-$ |
|---|---|---|---|---|---|---|---|
| SafetyHopperVelocity-v1 | 25 | 9 | - 9 | SafetyAntVelocity-v1 | 0.5 | 0.25 | -0.25 |
| SafetyHumanoidStandup-v4 | 1400 | 300 | 0 | SafetyWalker2d-v4 | 40 | $+\infty$ | 0 |
| SafetyWalker2d-v4-a | 30 | $+\infty$ | 0 | SafetyWalker2d-v4-b | 40 | $+\infty$ | 0 |
| SafetyReacher-v4 | 40 | 0 | $-\infty$ | | | | |

Table 6: Cost limit and slack parameters used in *Omnisafe* and *Safety-MuJoCo* experiments. The results of *SafetyHopperVelocity-v1* and *SafetyAntVelocity-v1* are shown in Figure 3, the results of *SafetyHumanoidStandup-v4* and *SafetyWalker2d-v4* are shown in Figure 2, the results of *SafetyWalker2d-v4-a* and *SafetyWalker2d-v4-a* are shown in Figures 4 (a), (b) and (c), the results of *SafetyWalker2d-v4-b* are shown in Figures 4 (d), (e) and (f); the results of *SafetyReacher-v4* experiments are shown in Figure 2.

| values \ algorithms / parameters | CUP | PCPO | PPOLag | ESPO |
|---|---|---|---|---|
| device | cpu | cpu | cpu | cpu |
| torch threads | 1 | 1 | 1 | 1 |
| vector env nums | 1 | 1 | 1 | 1 |
| parallel | 1 | 1 | 1 | 1 |
| epochs | 500 | 500 | 500 | 500 |
| steps per epoch | 20000 | 20000 | 20000 | \ |
| update iters | 40 | 10 | 40 | 10 |
| batch size | 64 | 128 | 64 | 128 |
| target kl | 0.01 | 0.01 | 0.02 | 0.01 |
| entropy coef | 0 | 0 | 0 | 0 |
| reward normalize | False | False | False | False |
| cost normalize | False | False | False | False |
| obs normalize | True | True | True | True |
| use max grad norm | True | True | True | True |
| max grad norm | 40 | 40 | 40 | 40 |
| use critic norm | True | True | True | True |
| critic norm coef | 0.001 | 0.001 | 0.001 | 0.001 |
| gamma | 0.99 | 0.99 | 0.99 | 0.99 |
| cost gamma | 0.99 | 0.99 | 0.99 | 0.99 |
| lam | 0.95 | 0.95 | 0.95 | 0.95 |
| lam c | 0.95 | 0.95 | 0.95 | 0.95 |
| clip | 0.2 | \ | 0.2 | \ |
| adv estimation method | gae | gae | gae | gae |
| standardized rew adv | True | True | True | True |
| standardized cost adv | True | True | True | True |
| cg damping | \ | 0.1 | \ | 0.1 |
| cg iters | \ | 15 | \ | 15 |
| hidden sizes | [64, 64] | [64, 64] | [64, 64] | [64, 64] |
| activation | tanh | tanh | tanh | tanh |
| lr | 0.0003 | 0.001 | 0.0003 | 0.001 |
| lagrangian multiplier init | 0.001 | \ | 0.001 | \ |
| lambda lr | 0.035 | \ | 0.035 | \ |

Table 7: Key hyparameters used in *Omnisafe* experiments. In ESPO, the steps of each epoch is determined by Algorithm 1, with Equations (7) and (6), in which the $X$ is 20000. The parameters for the baselines are consistent with those of *Omnisafe*, and their performance is meticulously fine-tuned in *Omnisafe* [32].

# E Impact and Limitation Statements

**Impact Statements:**   This paper presents work aiming to advance the field of safe RL, and we believe the study can significantly benefit multi-objective optimization efficiency, such as optimizations with ten or even hundreds of objectives. Additionally, this paper shares the general societal impact of progress in this domain, including both potential positive applications and negative consequences such as misuse. As capabilities in safe RL advance toward real-world deployment, continued monitoring and assessment of broader impacts remain warranted, as well as ensuring ethical deployment.

**Limitation Statements:**   The study currently focuses on simulation experiments; however, in the future, we aim to deploy our method in real-world applications. Additionally, although it is necessary to set sample size parameters for policy optimization, our method demonstrates superior performance across multiple tasks compared to SOTA baselines. Notably, our method's performance won't be heavily sensitive to the hyperparameters from our observations, as observed in our analysis (refer to Section 5.3). It is important to acknowledge that there is no free launch, and an algorithm could not address everything[5].

---

[5] https://locall.host/is-there-an-algorithm-for-everything/

