# OpenReview forum: "Enhancing Efficiency of Safe Reinforcement Learning via Sample Manipulation"
_NeurIPS.cc/2024/Conference — NeurIPS 2024 poster_

### Official Review · Reviewer_qrLu · 2024-06-19

**Soundness:** 3
**Presentation:** 2
**Contribution:** 2
**Rating:** 5
**Confidence:** 3

**Summary:**

This paper proposes a safe RL algorithm that adjusts the sample number during one on-policy update based on the conflict between the reward gradient and the cost gradient. When the policy's cost value is near the constraint threshold and there is a gradient conflict, a larger sample number is used; otherwise, a smaller sample number is applied. The paper also provides theoretical results about convergence rates, reducing oscillation, and sample efficiency. Finally, the paper evaluates the algorithm on two safe RL benchmarks.

**Strengths:**

- It is an interesting and innovative attempt to improve sample efficiency for safe RL by dynamically adjusting the sample number of on-policy updates based on gradient conflict.
- Some efforts on theoretical analysis are provided to justify the proposed approach.
- Empirical results demonstrate the capability of this work to improve performance and sample efficiency.

**Weaknesses:**

- The sample manipulation is an interesting mechanism, and the gradient conflict, as a manipulation signal, occurs widely in many safe RL algorithms, including both primal and primal-dual methods. However, it is solely applied to PCRPO, which may weaken its generalization and persuasiveness and makes it seem like a minor improvement tailored to PCRPO.
- The description of the main backbone PCRPO is insufficient, making it difficult to understand the whole algorithm pipeline without prior knowledge of PCRPO. See question 1, 2.
- Some points about the theoretical results remain to be clarified. See questions 3 and 4.
- Some experimental settings are debatable. See question 5.

**Questions:**

1. Eq. (3)(4) of PCRPO, serving as an essential component of your algorithmic pipeline, is kind of indigestible.  How to set $x^r_t$, $x^c_t$? And how is Eq. (3) functioning as 'projecting reward and cost gradients onto their normal planes'? It would be better to add more details or illustrations about PCRPO (at least in the appendix).
2. I see the $h^+,h^-$ dynamically change in rows 4, 7 in Algorithm 1 ESPO. Is it an existing component of PCRPO or a new trick in ESPO?
3. From the description of Theorem 4.1 and Proposition 4.2, I cannot find the association between sample manipulation and these two theoretical results. Can I view them as conclusions about PCRPO and not tailored to ESPO?
4. It seems that Assumption A.7, which bridges sample size and performance, serves as a very important theoretical base to verify the sample size manipulation. There should be more intuitive explanations and discussions to justify this assumption.
5. I guess you use a fixed number of training epochs for all algorithms. But training epochs are not a good metric for comparison, especially when each epoch corresponds to different sample steps, so maybe the x-axis of Figures 2 and 3 should be set to sample steps rather than training epochs to fairly demonstrate the sample efficiency of all algorithms. Besides, Table 1 should report the sample steps used to reach the same performance rather than the sample steps in a fixed number of epochs.

**Limitations:**

The limitations are discussed in the paper.

---

> ### Author Rebuttal · Authors · 2024-08-07
>
> # Reply to Reviewer qrLu
>
> We appreciate the reviewer for recognizing our contributions in both practice and theory, and for providing constructive suggestions.
>
> > **Q1:** The sample manipulation is an interesting mechanism with the gradient conflict as a signal.  Is it solely applied to PCRPO or can be extended to other safe RL algorithms?
>
>
> **A1:** Many thanks to the reviewer's recognition of the potential generalization of our sample manipulation approach. The reviewer is insightful that using the gradient conflict as a manipulation signal (metric) can improve the sample efficiency of extensive safe RL algorithms, not only PCRPO. Please refer to **Q1** in the general response.
>
>
>
> > **Q2:** How to set $x^r_t$ or $x^c_t$ ? Explain Eq. (3)(4) of PCRPO and how is Eq. (3) functioning as 'projecting reward and cost gradients onto their normal planes'? Add more details about PCRPO.
>
> **A2:** Thank the reviewer for the clarification suggestions.
>
> - $x_t^r$ and $x_t^c$ represent the weights of the reward gradient and the safety cost gradient in the final gradient $w_{t+1}$. Please refer to **Q2** of the general response for more details, where we explain how to choose/set $x_t^r,x_t^c$ and also **conduct new ablation studies using varying $x_t^r$ and $x_t^c$**.
> - As the reviewer suggested, we have added an intuitive description for Eq.(3) in the main text and a more detailed introduction and illustration for PCRPO in the appendix. Recall the first line of Eq.(3) serves as the update rule when there is some conflict between the reward gradient and the safety cost gradient. In that case, to explain "project the reward and cost gradients onto their normal planes", we take the first term of Eq.(3) $g_r  - \frac{g_r \cdot g_c}{ ||g_r||^2} g_c$ as an example. $\frac{g_r \cdot g_c}{ ||g_r||^2} g_c$ represent the vector projection of reward gradient $g_r$ on the cost gradient $g_c$. And the corresponding surrogate reward gradient $g_r - \frac{g_r \cdot g_c}{ ||g_r||^2} g_c$ becomes a vector that is perpendicular to $g_c$ --- the surrogate reward gradient not only improves reward but also won’t conflict with the cost gradient $g_c$ (since it is perpendicular to $g_c$, which we called the normal plane).
>
>
> > **Q3:** I see the dynamically change in rows 4, 7 in Algorithm 1 ESPO. Is it an existing component of PCRPO or a new trick in ESPO?
>
> **A3:** We apologize for this confusion. The dynamic change in rows 4 and 7 of Algorithm 1 (ESPO) is indeed an existing component from PCRPO, not a new trick introduced in ESPO. This mechanism in PCRPO is designed to dynamically adjust the range of soft region (by setting $h^+, h^-$) to adapt the optimization behaviors. This trick can improve the trade-offs between reward performance and safety constraints objectives, which we maintained in ESPO to leverage its efficiency.
>
> The contributions of ESPO focus on developing the new sample manipulation paradigm using gradient conflict as the metric/signal, which shows powerful advantages in sample efficiency, as shown in the paper and our new experiments (please refer to the previous answers **A1** to Q1).
>
> > **Q4:** Are the theoretical results in Theorem 4.1 and Proposition 4.2 conclusions about PCRPO but not tailored to ESPO?
> - **A4:** This is indeed one of the essential theoretical contributions of this work. The reviewer is correct that the convergence guarantees (Theorem 4.1) and the advantages of reduced optimization oscillation (Proposition 4.2) for our ESPO also hold for the prior work PCRPO, which are direct implications of our results. Notably, PCRPO is currently one of the state-of-the art safe RL algorithms, while still needing theoretical guarantees. This work closes this gap for PCRPO. In addition, ESPO is a broader algorithm framework  --- PCRPO can be seen as a special case with identical sample size choices for all training iterations, which all the theoretical results hold for.
>
> > **Q5:** Assumption A.7 serves as a key to verify the sample size manipulation. Give more intuitive explanations and discussions to justify this assumption.
> - **A5:** We have added discussions to justify Assumption A.7 in the new version: In words, Assumption A.7 assumes a local Lipschitz property of the Q function error term $\delta_t = \|Q_r^{\pi_{w_t}}  - \overline{Q}_t^r\|_2$ with respect to the sample size $s_t^B$ used for estimating $\overline{Q}_t^r$ at any time step $t$. Typically, this can be satisfied easily since $\delta_t$ generally will decrease monotonically as the sample size increases in a polynomial order, such as $\delta_t \approx O(\sqrt{\frac{1}{s_t^B}})$ that usually holds with high probability verified by high-dimensional statistics (some concentration inequalities).
>
>
> > **Q6:** Change the x-axis of Figures 2 and 3 to sample steps rather than training epochs for fair comparisons. Besides, Table 1 should report the sample steps used to reach the same performance rather than that used in a fixed number of epochs.
>
> **A6:** As the reviewer suggested, to provide a more fair and informative evaluation for the proposed method, we have made the following changes to show the results:
>
> - For Figures 2 and 3: We have revised them to use the number of sample steps as the x-axis rather than training epochs, shown in **Figures 5 and 6 in the pdf of the general response**. This more reasonable illustration further underscores the sample efficiency of our proposed ESPO compared to baselines.
> For both Table 1 and Table 2: We have revised them (in the general response pdf file Table 1 and Table 2) to report the number of sample steps required to reach reasonable performance thresholds, rather than after a fixed number of epochs.
> - Using these more fair metrics that the reviewer suggested, the results demonstrate even more efficiency of our ESPO compared to the baselines, in terms of not only sample efficiency, but also reward and safety performance.

---

> ### Comment · Reviewer_qrLu · 2024-08-08
>
> Thanks for the authors' response. The supplementary experimental results are comprehensive, and the additional clarifications address most of my confusions. Thus, I decide to raise the score to 4. However, I still hold a slightly negative opinion due to two main concerns: (1) the limited applicability of sample manipulation to broader algorithms (the paper structure seems overly tailored to PCRPO, even though some additional experimental results about TRPO-Lag and CRPO are provided), and (2) the weak connection between the theoretical analysis and the main contribution (sample manipulation).
>
> Just a reminder, it seems there is a typo in your response A2 where $g_r-\frac{g_r\cdot g_c}{||g_r||^2}g_c$ should be corrected to $g_r-\frac{g_r\cdot g_c}{||g_c||^2}g_c$ according to the paper.

---

> > ### Author Response · Authors · 2024-08-08
> >
> > Dear Reviewer,
> > Thank you for engaging in discussion with us and for appreciating our new experimental results! We will discuss on your further insightful comments. We have corrected the typo to $g_r - \frac{g_r - g_c}{\|g_c\|^2}g_c$. Regarding your other comments:
> > - **Adapting the presentation to highlight the generalization of sample manipulation to extensive safe RL algorithms and problems.** The reviewer is correct that we did a step-by-step presentation to introduce the proposed algorithm ESPO, it is indeed built on PCRPO which serves as an example in our algoirthm framework. As the reviewer suggested, we have decided to adapt the presentation for further clarification (in section 4 and appendix) by following the structure as below:
> >     - **Showing the key module sample manipulation** --- depending on the conflict signal of reward and cost gradients.
> >     - **How to integrate it into diverse safe RL algorithms**: introducing PCRPO and then the corresponding ESPO (PCRPO + sample manipulation) as a detailed example; introduce other examples regarding the primal method CRPO and primal-dual method TRPO-Lagrangian. (The experimental results are presented in Section 5 and **Q1** in the general response.)
> >     - **How to integrate it into more complex safe RL problems** --- multi-objective safe RL problems.  (The experimental results are presented in **Q1** in the general response.)
> >
> > We appreciate the reviewer for enabling us to verify the generalization power of our sample manipulations and helping us improve the current paper organization.
> >
> >
> >
> > - **Main contributions on the theoretical side:** Thanks for raising this question, which is indeed an essential contribution in our theoretical part. A brief answer would be: our theoretical analysis is a more general framework that is not only for sample manipulation module, but extensive primal-based safe RL algorithms. The separation of the theoretical framework from sample manipulation is not a flaw, but an intentional advantage for generalization.
> > Specifically, recall that we provide three provable advantages for ESPO --- 1) convergence in terms of both the optimal reward and the constraint requirement (Theorem 4.1); 2) efficient optimization with reduced oscillation (Proposiiton 4.2); sample efficiency with sample size manipulation (Propositon 4.3).
> >     - **The sample efficiency guarantee directly results from sample manipulation.** The reviewer is correct in noting that the provable advantages of ESPO also depend on other modules since all modules influence the optimization process. But sample manipulation plays a key role in supporting the sample efficiency guarantees for ESPO.
> >     - **The separation ability of the theoretical framework from sample manipulation is not a flaw, but an intentional advantage for generalization.** We would like to highlight that our theoretical guarantees (also the technical tools) for optimization perspective --- convergence (Theorem 4.1) and the advantages of reduced optimization oscillation (Proposition 4.2) --- can potentially work for a lot more primal-based safe RL algorithms even without sample manipulation (such as PCRPO). This is attributed to the fact that the theoretical analysis can be (mostly) decomposed into an optimization part and a statistical part, where sample manipulation primarily has an impact.  Notably, PCRPO is currently one of the state-of-the-art safe RL algorithms, while still needing theoretical guarantees for convergence. Our theoretical results for ESPO also hold for the prior work PCRPO, which are direct implications of our results. This work closes this gap for PCRPO and can be useful to provide convergence guarantees for extensive primal-based safe RL algorithms.

---

> > > ### Author Response · Authors · 2024-08-12
> > >
> > > Dear Reviewer,
> > >
> > > Thank you for your valuable comments. We have responded to your comments regarding the differences compared to PCRPO and our theoretical contributions. If you have any further questions or comments, please don't hesitate to let us know.
> > >
> > > As the rebuttal deadline is approaching, we hope our response can address your concerns. We appreciate your time and expertise in reviewing our work.
> > >
> > > With gratitude
> > >
> > > Authors

---

> > > > ### Comment · Reviewer_qrLu · 2024-08-12
> > > >
> > > > Thanks for your further explanations, I have raised my score.

---

> > > > > ### Author Response · Authors · 2024-08-13
> > > > >
> > > > > Dear Reviewer,
> > > > >
> > > > > Thank you for raising your score. We sincerely appreciate your constructive feedback, which is valuable to our work. We are grateful for your time and effort in reviewing our work.
> > > > >
> > > > > Best regards,
> > > > >
> > > > > The Authors

---

### Official Review · Reviewer_by8k · 2024-07-11

**Soundness:** 2
**Presentation:** 3
**Contribution:** 2
**Rating:** 5
**Confidence:** 4

**Summary:**

The paper introduces an approach, Efficient Safe Policy Optimization (ESPO), aimed to improve the efficiency of safe reinforcement learning (RL). ESPO tries to enhance sample efficiency through sample manipulation, addressing the challenges of sample inefficiency in safe RL, which often requires extensive interactions with the environment to learn a safe policy. The proposed method dynamically adjusts the sampling process based on the observed conflict between reward and safety gradients.

**Strengths:**

1. The paper is well motivated, articulates the challenges in existing safe RL methods and justifies the need for dynamic sample manipulation.

2. ESPO's dynamic sample manipulation based on gradient conflicts seems relevant to the field of safe RL, potentially reducing computational costs and improving learning efficiency.

3. The paper provides both theoretical analysis and empirical validation.

**Weaknesses:**

1. While the paper evaluates ESPO on two benchmarks, additional evaluations on more diverse and complex environments would strengthen the generalizability claims.

2. More in-depth comparisons with a broader range of SOTA methods could provide a clearer picture of ESPO's relative performance.

**Questions:**

1. How does ESPO perform in environments with high-dimensional state and action spaces compared to low-dimensional ones?

2. What is the impact of noisy gradient estimates on the performance and stability of ESPO?

3. Can ESPO be effectively integrated with off-policy or model-based RL methods to further enhance sample efficiency?

**Limitations:**

1. ESPO's reliance on gradient conflict signals for sample manipulation might limit its applicability in environments where gradient estimation is noisy or unreliable.

2. The method's scalability to very large or real-time environments is not thoroughly explored, raising questions about its practical deployment in such settings.

3. The sensitivity of ESPO to various hyperparameters, such as learning rates and sample size thresholds, needs further investigation.

---

> ### Author Rebuttal · Authors · 2024-08-07
>
> # Reply to Reviewer by8k
>
> Many thanks to the reviewer for recognizing our contributions in terms of theory and practice.
>
> > **Q1:** Adding evaluations on more diverse, complex, or real-time environments would strengthen the generalizability claims.
>
> **A1:** Thank the reviewer for insightful comments. We agree that testing on additional benchmarks or safe RL tasks is essential to substantiate the generalizability of ESPO. Following the reviewer's suggestion,
> - **We add three new sets of experiments on either a more complex class of safe RL problem --- multi-objective safe RL, or integrating our proposed technical modules with other safe RL algorithms to show the generalizability**. Please refer to **Q1** in the general response for details. The results from these additional experiments affirm that our method not only generalizes well to diverse, safe RL problems but can potentially bring benefits for extensive safe RL algorithms.
> - More experiments on large and real-time environments leave to future work: (1) extending ESPO to safe MARL tasks, which involve multi-agent strategic interactions while maintaining safety constraints with large-scale applications in collaborative robotics or autonomous systems. (2) Deploying ESPO in real-robot control tasks requires addressing real-world challenges such as sensor noise, actuator delays, and environmental uncertainties.
>
> > **Q2:** More in-depth comparisons with a broader range of SOTA methods
> - **A2:** Thank you for raising this point. First, we recall that we include the SOTA and classical ones of both primal methods (CRPO, PCRPO) and primal-dual methods (PCPO, CUP, PPOLag), which are the main classes of safe RL algorithms. So, instead of including more similar baselines, we choose to extend our ESPO to a new safe RL problem --- multi-objective safe RL and compare it to this problem's SOTA baseline CRMOPO. Details can be referred to the previous answer **A1**. These additional experiments verify the efficiency of ESPO (or its variants) not only in the standard safe RL problems, but also in more complex tasks.
>
> > **Q3:** How does ESPO perform in environments with high-dimensional state and action spaces compared to low-dimensional ones?
>
> **A3:** Thank you for your insightful question. We observe that the proposed method ESPO demonstrates superior performance over baselines in both low-dimensional tasks (SafetyHopperVelocity-v1 (action: 3D, state: 10D) and SafetyReacher-v4 (action: 2D, state: 10D)) and relatively high-dimensional tasks (SafetyAntVelocity-v1 (action: 8D, state: 27D), SafetyWalker-v4 (action: 6D, state: 17D), and SafetyHumanoidStandup-v4 (action: 17D, state: 45D). ), but with particularly strong performance in low-dimensional tasks. For details, please refer to Figures 5 and 6 in the general response file.
> - It is an interesting direction to implement ESPO for tasks with higher dimensional state and action, such as using an image as the state. This will involve more challenges, such as representation learning and computer vision. ESPO has potential in such high-dimensional tasks due to its sample efficiency advantages compared to prior art matches the pressing need of reducing required samples in those tasks -- such as the video game benchmark Atari (with the image as input) is typically one of the hardest benchmarks to solve and cost a lot of times to collect samples [1].
>
> [1] Ye, Weirui, et al. "Mastering atari games with limited data." NeurIPS 2021.
>
> > **Q4:** What is the impact of noisy gradient estimates on the performance and stability of ESPO due to its reliance on gradient conflict signals?
>
> **A4:** We appreciate the reviewer's insightful question. Typically, for all policy-based or actor-critic (safe) RL algorithms, the noisy gradient estimates can bring challenges since they are usually estimated with a batch of samples but not infinite samples.
>
> - **Inherent challenges for safe RL problems:** The reviewer is correct that the noisy gradient estimates bring challenges for ESPO, while it is not primarily brought by algorithm design, but the inherent daunting challenges for safe RL problems ---a good balanced update direction is hard to find using noisy reward gradient and safety cost gradient estimates (there may be conflict happens between reward and cost).
> - **The sample manipulation using gradient conflict signals (our key technical module in ESPO) can potentially reduce the effects of noisy gradient estimates.** Instead of limiting the applicability, the proposed sample manipulation is actually inspired by the noisy gradient estimate issue and designed to reduce its effect. In summary, the sample manipulation module will use more samples to estimate the final gradient when it needs more accuracy (when there is possibly a conflict between the reward and cost and a balance is required) and fewer samples when it tolerant more error --- improve sample efficiency. We acknowledge that the gradient conflict signal metric may also be noisy, but it won't directly hurt the final gradient direction. ESPO's superior performance implicitly shows that the performance is not sensitive to the gradient conflict signal errors.
>
> > **Q5:** Can ESPO be effectively integrated with off-policy or model-based RL methods?
> - **A5:** The answer is Yes. The core capability of ESPO lies in its use of sample manipulations based on reward and cost gradients to improve sample efficiency. This mechanism can be seamlessly incorporated into the frameworks of off-policy and model-based approaches, where gradients are readily available.  While it has not been tested, its fundamental principles suggest that such integration is feasible and potentially very beneficial. This is an exciting direction for future work.
>
> > **Q6** The sensitivity of ESPO to various hyperparameters, such as learning rates and sample size thresholds.
> - **A6:** Thank the reviewer for the comments on hyperparameters. Please refer to **Q3** in the general response for details.

---

> > ### Author Response · Authors · 2024-08-12
> >
> > Dear Reviewer,
> >
> > We sincerely appreciate your valuable comments on our paper. In response, we have deployed our method to more challenging tasks and other algorithms, including: (1) Primal-based methods (e.g., CRPO [1]); (2) Primal-dual based methods (e.g., TRPOLag [2,3]); (3) Safe multi-objective reinforcement learning (e.g., CRMOPO [4]). The results of these experiments demonstrate that our method exhibits superior performance compared to state-of-the-art baselines across diverse tasks and algorithms in terms of safety, reward, and sample efficiency. These findings suggest that our sample manipulation approach can serve as a general method for safe RL and potentially extend to multi-objective learning scenarios.
> >
> > If you have any further questions or require additional clarification, please don't hesitate to let us know. As the rebuttal deadline approaches, we hope our response can address your concerns, and we are grateful for your time and expertise in reviewing our work.
> >
> > With gratitude,
> >
> > The Authors
> >
> > > [1] Xu, T., Liang, Y., & Lan, G. (2021, July). Crpo: A new approach for safe reinforcement learning with convergence guarantee. In International Conference on Machine Learning (pp. 11480-11491). PMLR.
> > [2] Ray, A., Achiam, J., & Amodei, D. (2019). Benchmarking safe exploration in deep reinforcement learning. arXiv preprint arXiv:1910.01708, 7(1), 2.
> > [3] Ji, J., Zhou, J., Zhang, B., Dai, J., Pan, X., Sun, R., ... & Yang, Y. (2023). Omnisafe: An infrastructure for accelerating safe reinforcement learning research. arXiv preprint arXiv:2305.09304.
> > [4] Gu, S., Sel, B., Ding, Y., Wang, L., Lin, Q., Knoll, A., & Jin, M. (2024). Safe and Balanced: A Framework for Constrained Multi-Objective Reinforcement Learning. arXiv preprint arXiv:2405.16390.

---

### Official Review · Reviewer_oXwv · 2024-07-12

**Soundness:** 3
**Presentation:** 3
**Contribution:** 2
**Rating:** 6
**Confidence:** 3

**Summary:**

The paper introduces Efficient Safe Policy Optimization (ESPO), an approach that enhances safe reinforcement learning by dynamically adjusting sample sizes based on gradient conflicts. ESPO optimizes reward and safety, improves convergence stability, and reduces sample complexity. The experiments shows the proposed approach outperforms existing methods for both higher reward and improved safety with fewer samples and less training time.

**Strengths:**

1. The paper introduced Efficient Safe Policy Optimization (ESPO), which dynamically adjusts sample sizes based on observed conflicts between reward and safety gradients. The approach is novel
2. The paper provides a comprehensive theoretical analysis of ESPO, including convergence rates and optimization stability.
3. The paper conducted experiments on the Safety-MuJoCo and Omnisafe benchmarks, ESPO demonstrates significant improvements over existing primal-based and primal-dual-based methods.

**Weaknesses:**

The paper experimented on SafetyReacher-v4, SafetyWalker2d-v4, and SafetyHopper/AntVelocity. Those safety tasks does not test generalization, such as those with safety gym.

**Questions:**

None

**Limitations:**

The authors discussed limitations.

---

> ### Author Rebuttal · Authors · 2024-08-06
>
> # Reply to Reviewer oXwv
>
> > **Q1:** The paper experimented on SafetyReacher-v4, SafetyWalker2d-v4, and SafetyHopper/AntVelocity. Those safety tasks do not test generalization, such as those with safety gym.
>
> **A1:**  We appreciate the reviewer's insightful comments. To the best of our understanding of the reviewer's question, we clarify the relationship between the benchmarks ([Safety-MuJoCo](https://github.com/SafeRL-Lab/Safety-MuJoCo) and [Omnisafe](https://github.com/PKU-Alignment/omnisafe)) used in this paper and prior popular benchmark (safe gym). We have added this important clarification in the new version.
> - [Safety-MuJoCo](https://github.com/SafeRL-Lab/Safety-MuJoCo) is a new environment benchmark adapted from [MuJoCo](https://github.com/google-deepmind/mujoco) and [Omnisafe](https://github.com/PKU-Alignment/omnisafe) is an algorithm benchmark (does not include new safe RL environments) that support the environments in [Safety-Gymnasium](https://github.com/PKU-Alignment/safety-gymnasium). Here, Safety-Gymnasium is a popular suite, an evolved version of the original Safety-Gym, developed to support the latest gym API and maintain compatibility with ongoing research needs, given that [Safety-Gym](https://github.com/openai/safety-gym) is no longer actively maintained.
> - We use SafetyReacher-v4, SafetyWalker-v4, SafetyHumanoidStandup-v4 in [Safety-MuJoCo](https://github.com/SafeRL-Lab/Safety-MuJoCo). [Safety-Gymnasium](https://github.com/PKU-Alignment/safety-gymnasium) employs cost constraints such as velocity limits and rewards based on the robot’s speed. In contrast, [Safety-MuJoCo](https://github.com/SafeRL-Lab/Safety-MuJoCo) benchmarks enable more kinds of safety cost constraints, such as the robot’s health—monitoring falls and joint forces— which haven’t been included/emphasized in [Safety-Gymnasium](https://github.com/PKU-Alignment/safety-gymnasium). This provides a more comprehensive evaluation of the algorithms to generalize across more types/numbers of safety-critical constraints.
> - Other environments (SafetyHopper/AntVelocity) that we used in this paper are exactly from [Safety-Gymnasium](https://github.com/PKU-Alignment/safety-gymnasium). Omnisafe mainly serves as an algorithm platform that is used to test safe RL baselines in environments from Safety-Gymnasium.

---

> ### Comment · Reviewer_oXwv · 2024-08-07
>
> Apologies for the confusion.
>
> Notice that the experiments conducted in the paper mainly use velocity as the safety constraint and is a fixed threshold.
> In the original safety gym environment (openai/safety-gym), the safety constraints are typically obstacles and hazards (e.g., a car reaching the goal without hitting obstacles/overlapping with hazards). The location of the obstacles/hazards varies in each run, thus testing generalization in some sense.
>
> I felt the safety-velocity benchmarks are easier than the openai safety-gym original settings (commonly used in SafeRL community). However, I acknowledge this is a weakness but not a solid rejection reason.

---

> > ### Author Response · Authors · 2024-08-08
> >
> > Thank you for engaging in the discussion and providing further clarification of the question. We really appreciate the reviewer for proposing this constructive suggestion to improve the quality of our work, making the generalization ability of ESPO more convincing and clearer. We provide **new experiments** as well as more dicussions. The results show the generalization ability of the proposed method ESPO to not only varying unsafe factors, but also diverse types of safety constraints.
> > - **New experiments on SafetyCarGoal1-v0 and SafetyPointGoal1-v0: ESPO has generalization ability to handle varying unsafe factors**. As the reviewer suggested, we conduct experiments on two new benchmarks SafetyCarGoal1-v0 and SafetyPointGoal1-v0: the car or the robot ball needs to navigate to the Goal’s location while circumventing Hazards that could vary in each running time. Hazards bring risks that could result in costs when an agent enters unsafe regions [5].
> > We follow the same experimental settings in this paper, The results show the required number of training steps when they reach the desired performance (reward) and satisfy the safety constraints, which show the superior sample efficiency of the proposed method ESPO.
> > - **The proposed algorithm ESPO also has the generalization ability to handle diverse types of safety constraints and a combination of them.** Notably, we would also like to highlight that we also conducted experiments on safe RL with constraints not only on velocities (SafetyHopperVelocity-v1 and SafetyAntVelocity-v1), but also others such as robot’s control force energy (SafetyReacher-v4, SafetyWalker-v4, SafetyHumanoidStandup-v4, see Section 5.1). ESPO showed superior performance on not only sample efficiency, but also reward perforamnce and safety satisfication.
> >
> >
> > **Table 1: Comparisons of the required sample steps for achieving the same desired reward while ensuring safety (cost limit: 15) on SafetyCarGoal1-v0 and SafetyPointGoal1-v0.**
> > | Task \ Algorithm | ESPO (Ours) | CUP [1] | PPOLag [2,3] | PCPO [4]|
> > |-----------------------------------|-------------|------|-------|---|
> > | SafetyCarGoal1-v0 (Reward:6.6)       | 1.9 M     | 4+ M         | 2.4 M  | 2.3 M|
> > | SafetyPointGoal1-v0 (Reward:3.7)       | 0.7 M     | 1.1 M         | 4+ M  | 1.7 M|
> >
> >
> >
> > > [1] Yang, L., Ji, J., Dai, J., Zhang, L., Zhou, B., Li, P., ... & Pan, G. (2022). Constrained update projection approach to safe policy optimization. Advances in Neural Information Processing Systems, 35, 9111-9124.
> > [2] Ji, J., Zhou, J., Zhang, B., Dai, J., Pan, X., Sun, R., ... & Yang, Y. (2023). Omnisafe: An infrastructure for accelerating safe reinforcement learning research. arXiv preprint arXiv:2305.09304.
> > [3] Ray, A., Achiam, J., & Amodei, D. (2019). Benchmarking safe exploration in deep reinforcement learning. arXiv preprint arXiv:1910.01708, 7(1), 2.
> > [4] Yang, T. Y., Rosca, J., Narasimhan, K., & Ramadge, P. J. Projection-Based Constrained Policy Optimization. In International Conference on Learning Representations, 2020.
> > [5] Ji, J., Zhang, B., Zhou, J., Pan, X., Huang, W., Sun, R., ... & Yang, Y. (2023). Safety gymnasium: A unified safe reinforcement learning benchmark. Advances in Neural Information Processing Systems, 36.

---

> > > ### Author Response · Authors · 2024-08-12
> > >
> > > Dear Reviewer,
> > >
> > > Thank you for your constructive comments. In response, we have deployed our method to Safety-Gym tasks to demonstrate its generalizability, including experiments on safe robot and car navigation in obstacle/hazard environments (SafetyCarGoal1-v0 and SafetyPointGoal1-v0 tasks).
> > >
> > > If you have any further questions or comments, please don't hesitate to let us know. As the rebuttal deadline approaches, we hope our response adequately addresses your concerns.
> > >
> > > We sincerely appreciate your time and expertise in reviewing our work.
> > >
> > > With gratitude,
> > >
> > > The Authors

---

> > > > ### Comment · Reviewer_oXwv · 2024-08-12
> > > >
> > > > Thank you, I have raised my score. However, my confidence is not high.

---

> > > > > ### Author Response · Authors · 2024-08-12
> > > > >
> > > > > Dear Reviewer,
> > > > >
> > > > > Thank you for raising your score. Your constructive comments are valuable to our work.
> > > > >
> > > > > We sincerely appreciate your thoughtful review and the effort you put into reviewing our work.
> > > > >
> > > > > Best regards,
> > > > >
> > > > > The Authors

---

### Official Review · Reviewer_qn1v · 2024-07-12

**Soundness:** 3
**Presentation:** 3
**Contribution:** 3
**Rating:** 6
**Confidence:** 4

**Summary:**

This paper presents a novel algorithm for safe reinforcement learning, ESPO,
which independently collects gradient information for reward optimization and
constraint satisfaction. It then makes a dynamic choice about how to combine
these two gradients in order to find an optimal safe policy. In addition, the
proposed algorithm dynamically adjusts the number of samples used for each
gradient computation resulting in a more sample-efficient learning process than
existing techniques. Theoretical results show that ESPO achieves near-optimal
reward and constraint satisfaction with high probability. Additional results
show that ESPO spends more time in safe regions during training than prior work.
In experiments, ESPO is able to achieve comparable or better reward and safety
behavior to prior approaches while requiring less training time.

**Strengths:**

- Safe RL is quite an important area, and advancements in efficient safe RL make
  it more applicable to real-world scenarios.
- The proposed algorithm is quite intuitive and handles a tricky issue
  (oscillation) appearing in other safe RL algorithms
- The algorithm is well-grounded in theory with Theorem 4.1. I also appreciate
  4.2, showing the improved constraint satisfaction at training time compared to
  existing work.
- The experimental results are promising. ESPO achieves comparable or better
  results in terms of both reward performance and constraint satisfaction to
  existing work, but uses fewer samples.

**Weaknesses:**

- There is minimal discussion of the coefficients $x_t^r$ and $x_t^c$ even
  though these seem like they should be critical to the algorithm's performance.
- It seems that the hyperparameters of ESPO require careful tuning which likely
  inhibits the deployment of this algorithm in practice. (I'm looking at Tables
  5 and 6 for this claim which show that the hyperparameters are quite different
  for different benchmarks.)
- In some cases the either $h^-$ or $h^+$ is infinite, meaning the algorithm
  never performs a pure constraint satisfaction step or a pure reward
  optimization step.

**Questions:**

- How are the coefficients $x_t^r$ and $x_t^c$ computed? What impact do they
  have on performance?
- How are the other hyperparameters ($\chi^+$, $\chi^-$, $h^+$, $h^-$) chosen?
  Does it require careful tuning for each environment?

**Limitations:**

The authors adequately discuss limitations.

---

> ### Author Rebuttal · Authors · 2024-08-06
>
> # Reply to Reviewer qn1v
> > **Q1:** More discussion of the critical coefficients $x_t^r$ and $x_t^c$ of the algorithm's performance. How are the coefficients $x_t^r$ and $x_t^c$ computed?
>
> **A1:** Thanks for raising this point! We conduct new ablation experiments and have added a detailed discussion for $x_t^r$ and $x_t^c$ in the algorithm design section with an intuitive explanation. Please refer to **Q2** in the general response.
>
>
> > **Q2:** Do the hyperparameters of ESPO require careful tuning? (Tables 5 and 6 use different hyperparameters for different benchmarks.)
>
> **A2:** Thanks for raising this point; we appreciate the opportunity to clarify it.
>
>
> - Tuning hyperparameters for different tasks is not unique to ESPO, but widely considered in designing safe RL algorithms. For instance, PCPO [1] requires fine-tuning the projection distance $D$ within a primal-dual safe RL framework. CUP [2] involves fine-tuning the safety hyper-parameter $v$, and FOCOPS [3] necessitates fine-tuning the safety temperature $\lambda$. For practical considerations, hyperparameter tuning is somewhat necessary for heavily distinct tasks, while it's often a one-time process for each new kind of environments.
> - Table 5 and Table 6 show the choices of the hyperparameters for different tasks: sample size adjustment parameters ($\zeta^+, \zeta^-$ defined in Equation (6-7)), safety constraint threshold (limit $b$), and soft region threshold parameters ($h^+, h^-$). **We indeed provide an ablation study in Figure 4 in Appendix C lines 688-709**, over the benchmark SafetyWalker2d. Specifically, Figures 4(a)-\(c) show the results of ESPO when the safety constraint threshold (limit $b$) is different (30 or 40) with comparisons to baseline CRPO, which exhibits similar and stable reward performance when the limit $b$ varies. In addition, Figures 4(d)-(f) show the ablation w.r.t. sample size adjustment parameters ($\zeta^+, \zeta^-$) and demonstrate the stability of the performance under different ($\zeta^+, \zeta^-$) settings. We acknowledge the need of reasonable fine-tuning hyperparameters and are actively working on adaptive techniques in future work.
>
>
>
> - > [1] Yang, T., et al. (2020). Projection-Based Constrained Policy Optimization. ICLR 2020.
> > [2] Yang, L., et al. (2022). Constrained update projection approach to safe policy optimization. NeurIPS 2022.
> > [3] Zhang, Y., et al. (2020). First order constrained optimization in policy space. NeurIPS 2020.
>
> > **Q3:** In some cases either $h^-$ or $h^+$ is infinite, meaning the algorithm never performs a pure constraint satisfaction step or a pure reward optimization step. How are they chosen and do they require tuning for each environment?
>
> **A3:** Our ESPO algorithm framework intentionally designs and allows for such choices. Different tasks have different preferences for optimizing rewards or prioritizing safety constraints. So generally, we need to choose/tune diverse $h^+, h^-$ to both satisfy the goal of different tasks and also the efficiency of the learning process, with intuitions shown below:
>
> - We can choose $h^- = -\infty$ (i.e., always consider the safety constraint objective even if constraints are already satisfied). Such a choice will potentially make less oscillation around the safety constraint requirement threshold and always update the reward objective and safety constraint objective together. So this choice fits the tasks that the agent also wants to keep safe during the learning process, but not only exploiting the process, or less oscillation of the optimization process, which can accelerate the learning process.
> - We can choose $h^+ =  \infty$ (i.e., always consider the reward objective even if constraints are violated). This choice is suitable for those tasks that have less priority for satisfying the safety constraint, or less oscillation of the optimization process can accelerate the learning process.
> - When $h^-, h^+$ are finite (the common choice): creating a soft region $[h^- +b, h^+ +b]$. When the safety cost objective falls into this soft region around the required limit $b$, both the reward objective and the safety cost objective will be considered for the optimization update rule, which can reduce the oscillation around the safety limit to some extent. With such finite choices, the algorithm keeps the possibility of optimizing only the reward or safety cost objective when the safety objective is pretty good or is largely violated, respectively.
>
> > **Q4:** How are the other hyperparameters ($X^+$, $X^-$, $h^-$, $h^+$) chosen? Does it require careful tuning for each environment?
>
> **A4:** The reviewer raises an important point about hyperparameter selections. To the best of the author's understanding, the reviewer is asking about how to choose the sample size adjustment parameters ($\zeta^+, \zeta^-$ defined in Equation (6-7)), and soft region threshold parameters ($h^+, h^-$).
> - The intuition of choosing $h^-$, $h^+$ are provided in above answer **A3**. Note that different tasks have different preferences for optimizing rewards or prioritizing safety constraints. So generally, we need to choose/tune diverse $h^+, h^-$ to both satisfy the goal of different tasks and also the efficiency of the learning process.
> - The sample size adjustment parameters $\zeta^+$ and $\zeta^-$ determine the final sample size $X(1+\zeta^+)$ or $X(1+\zeta^-)$ in different scenarios (whether the gradient conflict occurs). As such, the final sample size typically is set close to the original sample size $X$. **We provide ablation study w.r.t. $\zeta^+, \zeta^-$ in Figure 4(d)-(f) in Appendix C lines 688-709**, over the benchmark SafetyWalker2d. Figures 4(d)-(f) demonstrate the stability of the performance under different ($\zeta^+, \zeta^-$) settings. It verifies that the ESPO is typically not sensitive to $\zeta^+, \zeta^-$ and just needs reasonable tuning.

---

> > ### Comment · Reviewer_qn1v · 2024-08-12
> >
> > Thank you for your response and I apologize for the error in my question. I did indeed mean to ask about $\zeta^+$ and $\zeta^-$ rather than $\chi^+$ and $\chi^-$. I will keep my score in favor of accepting the paper.

---

> > > ### Author Response · Authors · 2024-08-12
> > >
> > > Dear Reviewer,
> > >
> > > Thank you for your follow-up message and for clarifying the focus of your question. We are grateful for your favorable scoring towards our paper.
> > >
> > > Your insights are valuable to our work. Thank you once again for your thorough review and consideration.
> > >
> > > Best regards,
> > >
> > > The Authors

---

### Author Rebuttal · Authors · 2024-08-06

# General Response:

We thank the reviewers for their careful reading of the paper and their insightful and valuable feedback. Here, we provide **new experimental results** and discussions to answer some common questions raised by reviewers.

**We attached a pdf file to show the required new experimental results/updated figures** for all reviewers, listed below:
* Figure 1: Integrate our algorithmic techniques in primal-dual based safe RL.
* Figure 2: Integrate our algorithmic techniques in primal based safe RL.
* Figure 3: Integrate our algorithmic techniques in a more complex class of problems ---- safe multi-objective RL.
* Figure 4: Conduct new ablation experiments regarding hyperparameters of gradient update rules or learning rates.
* Figures 5 & 6: Update figures to use the number of sample steps as the x-axis rather than training epochs.

**Q1: Generalization of our key technical module in the algorithm (sample manipulation) to other safe RL algorithms or problems (Q1 of reviewer qrLu)/ Broader evaluation of the proposed method ESPO (Q1 of reviewer by8k).**


**A1:** We appreciate the reviewers' acknowledgment of the potential for our sample manipulation approach as a general method. Following the reviewers' suggestions, we conduct **three new sets of experiments** to demonstrate the generalization and advantages of our sample manipulation module (the key technical module in our algorithm) by integrating them with representative existing safe RL algorithms:
- **1) Integrating with primal-dual method --- [TRPO Lagrangian](https://cdn.openai.com/safexp-short.pdf) for standard safe RL problems:** see Figure 1 in the general response file.
- **2) Integrating with primal method --- [CRPO](https://proceedings.mlr.press/v139/xu21a/xu21a.pdf) for standard safe RL problems:** see Figure 2 in the general response file.
- **3) Adapting ESPO to a class of more complex safe RL problems --- multi-objective safe RL problems:** The results are shown in Figure 3 in the general response file. We adapt ESPO to the multi-objective safe RL problem (termed as ECRMOPO) and evaluate it on a [safe multi-objective RL benchmark](https://arxiv.org/pdf/2405.16390),  with comparison to the SOTA baseline [CRMOPO](https://arxiv.org/pdf/2405.16390).


Summing up the results from these experiments affirms that our sample manipulation method can improve the sample efficiency of diverse safe RL problems and also extensive safe RL algorithms (including both primal-based and primal-dual-based methods). Those results also serve as a broader and more comprehensive evaluation of our proposed sample manipulation approach. We will emphasize the broad power of the sample manipulation design in the revised manuscript.

**Q2: Questions for choosing and tuning hyperpaprameters: $x_t^r$ and $x_t^c$ of the optimization update rule.**

**A2:** We have added a detailed discussion for $x_t^r$ and $x_t^c$ in the algorithm design section and provide an intuitive explanation here:
- **How to choose $x_t^r, x_t^c$.** $x_t^r$ (resp. $x_t^c$) represents the weight of the reward gradient (resp. the safety cost gradient) in the final gradient $w_{t+1}$. So $x_t^r + x_t^c=1$ all the time and for instance, $x_t^r=1$ (resp.  $x_t^c=1$)  means we only use reward gradient (resp. safety cost gradient), and $x_t^r = x_t^c = 0.5$ means reward and safety cost objectives are considered equally important in the overall gradient. In general, $x_t^r$ and $x_t^c$ are hyperparameters in the framework that we can either pre-set as a fixed value or adaptively adjust during the running process as needed. For instance, we can set $x_t^r$ to be larger if we care more about the reward performance; otherwise, we set $x_t^c$ to be larger to enhance safety. Throughout our experiments, we just set $x_t^r = x_t^c = 0.5$ for simplicity, which also showed superior performance than prior arts (see Section 5).
- **As the reviewers suggested, we add an ablation study for the hyperparameter $x_t^r$ and $x_t^c$ to evaluate its effect on the performance, shown in Figures 4(a)-\(c\) in the general response pdf file.** To test the sensitivity of the performance w.r.t. the hyperparameters $x_t^r$ and $x_t^c$, we add experiments using other two combinations of $x_t^r$ and $x_t^c$ ($x_t^r=0.4, x_t^c=0.6$ or $x_t^r=0.6, x_t^c=0.4$). The results show that our proposed ESPO performs well using such different $x_t^r, x_t^c$ settings and is even better than using $x_t^r = x_t^c = 0.5$ (in our paper) sometimes.


> **Q3: Ablation studies for other parameters, such as learning rates and sample size thresholds.**
- **A3:** We appreciate the reviewers' valuable comments. We indeed have ablation studies in Appendix C (due to the limit of the main text), and we have **conducted more ablations on learning rates and gradient weights**, see Figure 4 in the general response file.
- **During rebuttal, we add ablation studies on key hyperparameters --- learning rates ($l_r$) and the weights of gradients ($x_t^r, x_t^c$), shown in Figure 4 in the general response file**, reveal that ESPO is robust to variations in learning rates. The results demonstrate that ESPO performs well under reasonably varying learning rates or weights of gradients.
- **Ablation study for sample size thresholds ($\zeta^+,\zeta^-$) are provided in Figure 4(d)-(f) in Appendix C lines 688-709 of the paper**, over the benchmark SafetyWalker2d. Figures 4(d)-(f) demonstrate the stability of the performance under different ($\zeta^+, \zeta^-$) settings. It verifies that the ESPO is typically not sensitive to $\zeta^+, \zeta^-$ and just needs reasonable tuning.

---

### Decision · Program_Chairs · 2024-09-25

**Decision:**

Accept (poster)

**Comment:**

This paper proposes a safe RL method called ESPO that leverages the gradient conflict to enhance sample efficiency during policy learning. The base algorithm is mainly based on PCRPO (reference [19]), which decouples the safe policy learning process into solely reward maximization, cost minimization, and another stage that simultaneously optimizes both reward and cost. The proposed method ESPO improves over PCRPO by leveraging the conflicts between the reward and cost gradient to dynamically adjust the sample size accordingly. By assigning more samples for situations with gradient conflicts (reward and safety are more challenging to balance for policy learning), and reducing the samples for situations with no conflicts, the proposed method can achieve better sample efficiency as compared to PCRPO. The paper also provides a detailed theoretical analysis (mostly based on the theoretical analysis in CRPO (reference [38])). The strengths and weaknesses are summarized below:

**Strengths:**

-	The idea of leveraging reward and cost gradient conflict to dynamically adjust sample size is interesting. The empirical results also demonstrate its effectiveness.
-	Some efforts on theoretical analysis are provided to justify the proposed approach.

**Weaknesses:**

-	The practical algorithm and theoretical analysis are mostly built upon existing works (PCRPO and CRPO). The new component introduced in this paper is a simple sample size adjustment scheme, which is somewhat incremental. The theoretical analyses are mostly borrowed from CRPO with some adaptation, which in my opinion is not sufficiently acknowledged in the paper. Also, I feel that the theoretical analysis still has some gaps to fully cover the practical algorithm.
-	As mentioned by one of the reviewers, the proposed method contains lots of hyperparameters, which can be tricky to tune in practice. The authors also used quite different hyperparameters for different benchmark tasks, which is not a very good practice. I suggest the authors present the results produced by a minimal set of hyperparameters in the final version.
-	Decoupling reward maximization and cost minimization in safe RL has also been explored in other existing online and offline safe RL works such as [1, 2], which are not adequately acknowledged in the paper. Also, more comparisons with SOTA methods should be conducted, as suggested by reviewers.
-	Lastly, more description of the backbone algorithm PCRPO also needs to be added to the main text to improve the readability of this paper.

The paper received generally positive feedback from the reviewers. Most of the concerns from reviewers have been addressed during the Author Rebuttal phase. I hope the authors can further improve the paper to address the remaining issues in their final version.